# PROBABILISTIC IMPLICIT SCENE COMPLETION

**Dongsu Zhang, Changwoon Choi, Inbum Park & Young Min Kim**
Department of Electrical and Computer Engineering, Seoul National University
96lives@snu.ac.kr, changwoon.choi00@gmail.com,
{inbum0215, youngmin.kim}@snu.ac.kr

## ABSTRACT

We propose a probabilistic shape completion method extended to the continuous geometry of large-scale 3D scenes. Real-world scans of 3D scenes suffer from a considerable amount of missing data cluttered with unsegmented objects. The problem of shape completion is inherently ill-posed, and high-quality result requires scalable solutions that consider multiple possible outcomes. We employ the Generative Cellular Automata that learns the multi-modal distribution and transform the formulation to process large-scale continuous geometry. The local continuous shape is incrementally generated as a sparse voxel embedding, which contains the latent code for each occupied cell. We formally derive that our training objective for the sparse voxel embedding maximizes the variational lower bound of the complete shape distribution and therefore our progressive generation constitutes a valid generative model. Experiments show that our model successfully generates diverse plausible scenes faithful to the input, especially when the input suffers from a significant amount of missing data. We also demonstrate that our approach outperforms deterministic models even in less ambiguous cases with a small amount of missing data, which infers that probabilistic formulation is crucial for high-quality geometry completion on input scans exhibiting any levels of completeness.

## 1 INTRODUCTION

High-quality 3D data can create realistic virtual 3D environments or provide crucial information to interact with the environment for robots or human users (Varley et al. (2017)). However, 3D data acquired from a real-world scan is often noisy and incomplete with irregular samples. The task of 3D shape completion aims to recover the complete surface geometry from the raw 3D scans. Shape completion is often formulated in a data-driven way using the prior distribution of 3D geometry, which often results in multiple plausible outcomes given incomplete and noisy observation. If one learns to regress a single shape out of multi-modal shape distribution, one is bound to lose fine details of the geometry and produce blurry outputs as noticed with general generative models (Goodfellow (2017)). If we extend the range of completion to the scale of scenes with multiple objects, the task becomes even more challenging with the memory and computation requirements for representing large-scale high resolution 3D shapes.

In this work, we present continuous Generative Cellular Automata (cGCA), which generates multiple continuous surfaces for 3D reconstruction. Our work builds on Generative Cellular Automata (GCA) (Zhang et al. (2021)), which produces diverse shapes by progressively growing the object surface from the immediate neighbors of the input shape. cGCA inherits the multi-modal and scalable generation of GCA, but overcomes the limitation of discrete voxel resolution producing high-quality continuous surfaces. Specifically, our model learns to generate diverse sparse voxels associated with their local latent codes, namely sparse voxel embedding, where each latent code encodes the deep implicit fields of continuous geometry near each of the occupied voxels (Chabra et al. (2020); Jiang et al. (2020)). Our training objective maximizes the variational lower bound for the log-likelihood of the surface distribution represented with sparse voxel embedding. The stochastic formulation is modified from the original GCA, and theoretically justified as a sound generative model.

We demonstrate that cGCA can faithfully generate multiple plausible solutions of shape completion even for large-scale scenes with a significant amount of missing data as shown in Figure 1. To the best of our knowledge, we are the first to tackle the challenging task of probabilistic scene completion,

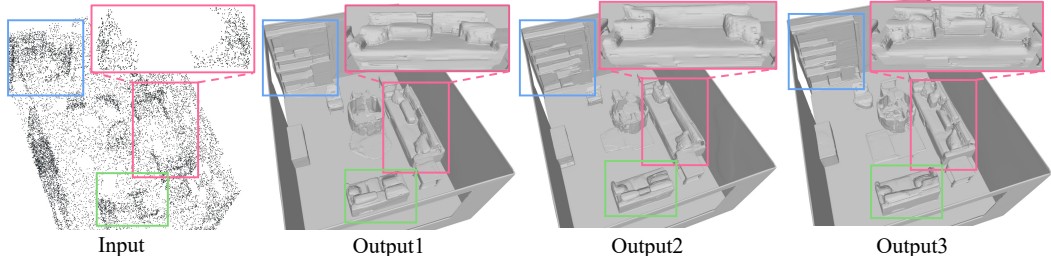

Figure 1: Three examples of complete shapes using cGCA given noisy partial input observation. Even when the raw input is severely damaged (left), cGCA can generate plausible yet diverse complete continuous shapes.

which requires not only the model to generate multiple plausible outcomes but also be scalable enough to capture the wide-range context of multiple objects.

We summarize the key contributions as follows: (1) We are the first to tackle the problem of probabilistic *scene* completion with partial scans, and provide a scalable model that can capture large-scale context of scenes. (2) We present continuous Generative Cellular Automata, a generative model that produces *diverse continuous surfaces* from a partial observation. (3) We modify infusion training (Bordes et al. (2017)) and prove that the formulation indeed increases the variational lower bound of data distribution, which verifies that the proposed progressive generation is a valid generative model.

## 2 PRELIMINARIES: GENERATIVE CELLULAR AUTOMATA

Continuous Generative Cellular Automata (cGCA) extends the idea of Generative Cellular Automata (GCA) by Zhang et al. (2021) but generates continuous surface with implicit representation instead of discrete voxel grid. For the completeness of discussion, we briefly review the formulation of GCA.

Starting from an incomplete voxelized shape, GCA progressively updates the local neighborhood of current occupied voxels to eventually generate a complete shape. In GCA, a shape is represented as a state $s = \{(c, o_c) | c \in \mathbb{Z}^3, o_c \in \{0, 1\}\}$, a set of binary occupancy $o_c$ for every cell $c \in \mathbb{Z}^3$, where the occupancy grid stores only the sparse cells on the surface. Given a state of an incomplete shape $s^0$, GCA evolves to the state of a complete shape $s^T$ by sampling $s^{1:T}$ from the Markov chain

$$s^{t+1} \sim p_\theta(\cdot \mid s^t), \tag{1}$$

where $T$ is a fixed number of transitions and $p_\theta$ is a homogeneous transition kernel parameterized by neural network parameters $\theta$. The transition kernel $p_\theta$ is implemented with sparse CNN (Graham et al. (2018); Choy et al. (2019)), which is a highly efficient neural network architecture that computes the convolution operation only on the occupied voxels.

The progressive generation of GCA confines the search space of each transition kernel at the immediate neighborhood of the current state. The occupancy probability within the neighborhood is regressed following Bernoulli distribution, and then the subsequent state is independently sampled for individual cells. With the restricted domain for probability estimation, the model is scalable to high resolution 3D voxel space. GCA shows that the series of local growth near sparse occupied cells can eventually complete the shape as a unified structure since the shapes are connected. While GCA is a scalable solution for generating diverse shapes, the grid representation for the 3D geometry inherently limits the resolution of the final shape.

## 3 CONTINUOUS GENERATIVE CELLULAR AUTOMATA

In Sec. 3.1, we formally introduce an extension of sparse occupancy voxels to represent continuous geometry named sparse voxel embedding, where each occupied voxel contains latent code representing local implicit fields. We train an autoencoder that can compress the implicit fields into the embeddings and vice versa. Then we present the sampling procedure of cGCA that generates 3D shape in Sec. 3.2, which is the inference step for shape completion. Sec. 3.3 shows the training objective of cGCA, which approximately maximizes the variational lower bound for the distribution of the complete continuous geometry.

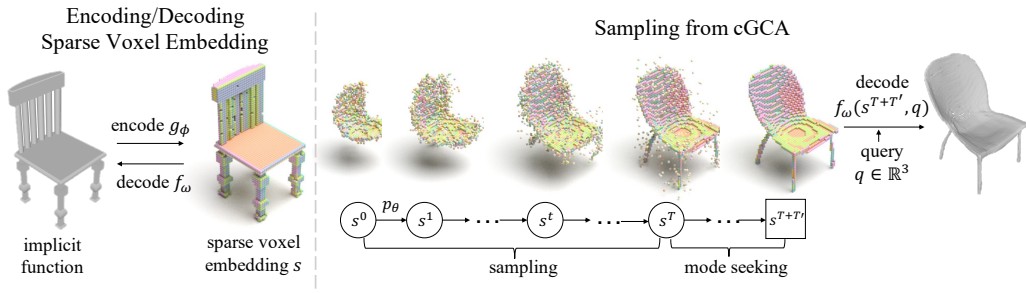

Figure 2: Overview of our method. The implicit function of continuous shape can be encoded as sparse voxel embedding $s$ and decoded back (left). The colors in the sparse voxel embedding represent the clustered labels of latent code $z_c$ for each cell $c$. The sampling procedure of cGCA (right) involves $T$ steps of sampling the stochastic transition kernel $p_\theta$, followed by $T'$ mode seeking steps which remove cells with low probability. From the final sparse voxel embedding $s^{T+T'}$, the decoder can recover the implicit representation for the complete continuous shape.

## 3.1 SPARSE VOXEL EMBEDDING

In addition to the sparse occupied voxels of GCA, the state of cGCA, named sparse voxel embedding, contains the associated latent code, which can be decoded into continuous surface. Formally, the state $s$ of cGCA is defined as a set of pair of binary occupancy $o_c$ and the latent code $z_c$, for the cells $c$ in a three-dimensional grid $\mathbb{Z}^3$

$$s = \{(c, o_c, z_c)|c \in \mathbb{Z}^3, o_c \in \{0, 1\}, z_c \in \mathbb{R}^K\}. \tag{2}$$

Similar to GCA, cGCA maintains the representation sparse by storing only the set of occupied voxels and their latent codes, and sets $z_c = 0$ if $o_c = 0$.

The sparse voxel embedding $s$ can be converted to and from the implicit representation of local geometry with neural networks, inspired by the work of Peng et al. (2020), Chabra et al. (2020), and Chibane et al. (2020a). We utilize the (signed) distance to the surface as the implicit representation, and use autoencoder for the conversion. The encoder $g_\phi$ produces the sparse voxel embedding $s$ from the coordinate-distance pairs $P = \{(p, d_p)|p \in \mathbb{R}^3, d_p \in \mathbb{R}\}$, where $p$ is a 3D coordinate and $d_p$ is the distance to the surface, $s = g_\phi(P)$. The decoder $f_\omega$, on the other hand, regresses the local implicit field value $d_q$ at the 3D position $q \in \mathbb{R}^3$ given the sparse voxel embedding $s$, $\hat{d}_q = f_\omega(s, q)$ for given coordinate-distance pairs $Q = \{(q, d_q)|q \in \mathbb{R}^3, d_q \in \mathbb{R}\}$. The detailed architecture of the autoencoder is described in Appendix A, where the decoder $f_\omega$ generates continuous geometry by interpolating hierarchical features extracted from sparse voxel embedding. An example of the conversion is presented on the left side of Fig. 2, where the color of the sparse voxel embedding represents clustered labels of the latent codes with k-means clustering (Hartigan & Wong (1979)). Note that the embedding of a similar local geometry, such as the seat of the chair, exhibits similar values of latent codes.

The parameters $\phi, \omega$ of the autoencoder are jointly optimized by minimizing the following loss function:

$$\mathcal{L}(\phi, \omega) = \frac{1}{|Q|} \sum_{(q,d_q) \in Q} |f_\omega(s, q) - \max\left(\min\left(\frac{d_q}{\epsilon}, 1\right), -1\right)| + \beta \frac{1}{|s|} \sum_{c \in s} \|z_c\|, \tag{3}$$

where $s = g_\phi(P)$ and $\epsilon$ is the size of a single voxel. The first term in Eq. (3) corresponds to minimizing the normalized distance and the second is the regularization term for the latent code weighted by hyperparameter $\beta$. Clamping the maximum distance makes the network focus on predicting accurate values at the vicinity of the surface (Park et al. (2019); Chibane et al. (2020b)).

## 3.2 SAMPLING FROM CONTINUOUS GENERATIVE CELLULAR AUTOMATA

The generation process of cGCA echos the formulation of GCA (Zhang et al. (2021)), and repeats $T$ steps of sampling from the transition kernel to progressively grow the shape. Each transition kernel

$p(s^{t+1}|s^t)$ is factorized into cells within the local neighborhood of the occupied cells of the current state, $\mathcal{N}(s^t) = \{c' \in \mathbb{Z}^3 \mid d(c, c') \leq r, c \in s^t\}$ [1] given a distance metric $d$ and the radius $r$:

$$p(s^{t+1}|s^t) = \prod_{c \in \mathcal{N}(s^t)} p_\theta(o_c, z_c|s^t) = \prod_{c \in \mathcal{N}(s^t)} p_\theta(o_c|s^t)p_\theta(z_c|s^t, o_c). \tag{4}$$

Note that the distribution is further decomposed into the occupancy $o_c$ and the latent code $z_c$, where we denote $o_c$ and $z_c$ as the random variable of occupancy and latent code for cell $c$ in state $s^{t+1}$. Therefore the shape is generated by progressively sampling the occupancy and the latent codes for the occupied voxels which are decoded and fused into a continuous geometry. The binary occupancy is represented with the Bernoulli distribution

$$p_\theta(o_c|s^t) = Ber(\lambda_{\theta,c}), \tag{5}$$

where $\lambda_{\theta,c} \in [0, 1]$ is the estimated occupancy probability at the corresponding cell $c$. With our sparse representation, the distribution of the latent codes is

$$p_\theta(z_c|s^t, o_c) = \begin{cases} \delta_0 & \text{if } o_c = 0 \\ N(\mu_{\theta,c}, \sigma^t \boldsymbol{I}) & \text{if } o_c = 1. \end{cases} \tag{6}$$

$\delta_0$ is a Dirac delta distribution at 0 indicating that $z_c = 0$ when $o_c = 0$. For the occupied voxels ($o_c = 1$), $z_c$ follows the normal distribution with the estimated mean of the latent code $\mu_{\theta,c} \in \mathbb{R}^K$ and the predefined standard deviation $\sigma^t \boldsymbol{I}$, where $\sigma^t$ decreases with respect to $t$.

**Initial State.** Given an incomplete point cloud, we set the initial state $s^0$ of the sampling chain by setting the occupancy $o_c$ to be 1 for the cells that contain point cloud and associating the occupied cells with a latent code sampled from the isotropic normal distribution. However, the input can better describe the provided partial geometry if we encode the latent code $z_c$ of the occupied cells with the encoder $g_\phi$. The final completion is more precise when all the transitions $p_\theta$ are conditioned with the initial state containing the encoded latent code. Further details are described in Appendix A.

**Mode Seeking.** While we effectively model the probabilistic distribution of multi-modal shapes, the final reconstruction needs to converge to a single coherent shape. Naïve sampling of the stochastic transition kernel in Eq. (4) can include noisy voxels with low-occupancy probability. As a simple trick, we augment *mode seeking steps* that determine the most probable mode of the current result instead of probabilistic sampling. Specifically, we run additional $T'$ steps of the transition kernel but we select the cells with probability higher than 0.5 and set the latent code as the mean of the distribution $\mu_{\theta,c}$. The mode seeking steps ensure that the final shape discovers the dominant mode that is closest to $s^T$ as depicted in Fig. 2, where it can be transformed into implicit function with the pretrained decoder $f_w$.

### 3.3 Training Continuous Generative Cellular Automata

We train a homogeneous transition kernel $p_\theta(s^{t+1}|s^t)$, whose repetitive applications eventually yield the samples that follow the learned distribution. However, the data contains only the initial $s^0$ and the ground truth state $x$, and we need to emulate the sequence for training. We adapt infusion training (Bordes et al. (2017)), which induces the intermediate transitions to converge to the desired complete state. To this end, we define a function $G_x(s)$ that finds the valid cells that are closest to the complete shape $x$ within the neighborhood of the current state $\mathcal{N}(s)$:

$$G_x(s) = \{\text{argmin}_{c \in \mathcal{N}(s)} d(c, c') \mid c' \in x\}. \tag{7}$$

Then, we define the infusion kernel $q^t$ factorized similarly as the sampling kernel in Eq. (4):

$$q_\theta^t(s^{t+1}|s^t, x) = \prod_{c \in \mathcal{N}(s^t)} q_\theta^t(o_c, z_c|s^t, x) = \prod_{c \in \mathcal{N}(s^t)} q_\theta^t(o_c|s^t, x)q_\theta^t(z_c|s^t, o_c, x). \tag{8}$$

The distributions for both $o_c$ and $z_c$ are gradually biased towards the ground truth final shape $x$ with the infusion rate $\alpha^t$, which increases linearly with respect to time step, i.e., $\alpha^t = \max(\alpha_1 t + \alpha_0, 1)$, with $\alpha_1 > 0$:

$$q_\theta^t(o_c|s^t, x) = Ber((1 - \alpha^t)\lambda_{\theta,c} + \alpha^t \mathbb{1}[c \in G_x(s^t)]), \tag{9}$$

---

[1] We use the notation $c \in s$ if $o_c = 1$ for $c \in \mathbb{Z}^3$ to denote occupied cells.

$$q_\theta^t(z_c|s^t, o_c, x) = \begin{cases} \delta_0 & \text{if } o_c = 0 \\ N((1 - \alpha^t)\mu_{\theta,c} + \alpha^t z_c^x, \ \sigma^t \boldsymbol{I}) & \text{if } o_c = 1. \end{cases} \quad (10)$$

Here $\mathbb{1}$ is an indicator function, and we will denote $o_c^x, z_c^x$ as the occupancy and latent code of the ground truth complete shape $x$ at coordinate $c$.

cGCA aims to optimize the log-likelihood of the ground truth sparse voxel embedding $\log p_\theta(x)$. However, since the direct optimization of the exact log-likelihood is intractable, we modify the variational lower bound using the derivation of diffusion-based models (Sohl-Dickstein et al. (2015)):

$$\log p_\theta(x) \geq \sum_{s^{0:T-1}} q_\theta(s^{0:T-1}|x) \log \frac{p_\theta(s^{0:T-1}, x)}{q_\theta(s^{0:T-1}|x)} \quad (11)$$

$$= \underbrace{\log \frac{p(s^0)}{q(s^0)}}_{\mathcal{L}_{\text{init}}} + \sum_{0 \leq t < T-1} \underbrace{-D_{KL}(q_\theta(s^{t+1}|s^t, x)\|p_\theta(s^{t+1}|s^t))}_{\mathcal{L}_t} + \underbrace{\mathbb{E}_{q_\theta}[\log p_\theta(x|s^{T-1})]}_{\mathcal{L}_{\text{final}}},$$

where the full derivation is in Appendix B.1. We now analyze $\mathcal{L}_{\text{init}}, \mathcal{L}_t, \mathcal{L}_{\text{final}}$ separately. We ignore the term $\mathcal{L}_{\text{init}}$ during optimization since it contains no trainable parameters. $\mathcal{L}_t$ for $0 \leq t < T - 1$ can be decomposed as the following :

$$\mathcal{L}_t = - \sum_{c \in \mathcal{N}(s^t)} \underbrace{D_{KL}(q_\theta(o_c|s^t, x)\|p_\theta(o_c|s^t))}_{\mathcal{L}_o}$$

$$+ q_\theta(o_c = 1|s^t, x) \underbrace{D_{KL}(q_\theta(z_c|s^t, x, o_c = 1)\|p_\theta(z_c|s^t, o_c = 1))}_{\mathcal{L}_z}, \quad (12)$$

where the full derivation is in Appendix B.2. Since $\mathcal{L}_o$ and $\mathcal{L}_z$ are the KL divergence between Bernoulli and normal distributions, respectively, $L_t$ can be written in a closed-form. In practice, the scale of $\mathcal{L}_z$ can be much larger than that of $\mathcal{L}_o$. This results in local minima in the gradient-based optimization and reduces the occupancy probability $q_\theta(o_c = 1|s^t, x)$ for every cell. So we balance the two losses by multiplying a hyperparameter $\gamma$ at $\mathcal{L}_z$, which is fixed as $\gamma = 0.01$ for all experiments.

The last term $\mathcal{L}_{\text{final}}$ can be written as following:

$$\mathcal{L}_{\text{final}} = \sum_{c \in \mathcal{N}(s^{T-1})} \log\{(1 - \lambda_{\theta,c})\mathbb{1}[o_c^x = 0]\delta_0(z_c^x) + \lambda_{\theta,c}\mathbb{1}[o_c^x = 1]N(z_c^x; \ \mu_{\theta,c}, \sigma^{T-1}I)\}. \quad (13)$$

A problem rises when computing $\mathcal{L}_{\text{final}}$, since the usage of Dirac distribution makes $\mathcal{L}_{\text{final}} \to \infty$ if $o_c^x = 0$, which does not produce a valid gradient for optimization. However, we can replace the loss $\mathcal{L}_{\text{final}}$ with a well-behaved loss $\mathcal{L}_t$ for $t = T - 1$ , by using the following proposition:

**Proposition 1.** *By replacing $\delta_0(z_c^x)$ with the indicator function $\mathbb{1}[z_c^x = 0]$ when computing $\mathcal{L}_{\text{final}}$, $\nabla \mathcal{L}_{T-1} = \nabla \mathcal{L}_{\text{final}}$, for $T \gg 1$*

*Proof.* The proof is found in Appendix B.3. □

The proposition above serves as a justification for approximating $\nabla \mathcal{L}_{\text{final}}$ as $\nabla \mathcal{L}_{T-1}$, with the benefits of having a simpler training procedure and easier implementation. Replacing $\delta_0(z_c^T)$ with $\mathbb{1}[z_c^x = 0]$ can be regarded as a reweighting technique that naturally avoids divergence in the lower bound, since both functions output a non-zero value only at $0$. Further discussions about the replacement are in Appendix B.3.

In conclusion, the training procedure is outlined as follows:

1. Sample $s^{0:T}$ by $s^0 \sim q^0$, $s^{t+1} \sim q_\theta^t(\cdot|s^t, x)$.
2. For $t < T$, update $\theta$ with $\theta \leftarrow \theta + \eta\nabla_\theta \mathcal{L}_t$, where $\eta$ is the learning rate.

Note that the original infusion training (Bordes et al. (2017)) also attempts to minimize the variational lower bound, employing the Monte Carlo approximation with reparameterization trick (Kingma & Welling (2014)) to compute the gradients. However, our objective avoids the approximations and can compute the exact lower bound for a single training step. The proposed simplification can be applied to infusion training with any data structure including images. We also summarize the difference of our formulation compared to GCA in Appendix C.

Table 1: Quantitative comparison of probabilistic scene completion in ShapeNet scene dataset with different levels of completeness. The best results are marked as bold. Both CD (quality, ↓) and TMD (diversity, ↑) in tables are multiplied by $10^4$.

| Method | min. rate 0.2 | | | min. rate 0.5 | | | min. rate 0.8 | | |
|---|---|---|---|---|---|---|---|---|---|
| | min. CD | avg. CD | TMD | min. CD | avg. CD | TMD | min. CD | avg. CD | TMD |
| ConvOcc | 3.60 | - | - | 1.33 | - | - | 0.74 | - | - |
| IFNet | 12.94 | - | - | 8.55 | - | - | 7.49 | - | - |
| GCA | 4.97 | 6.32 | **11.56** | 3.02 | 3.54 | **4.92** | 2.50 | 2.64 | 2.76 |
| cGCA | 2.80 | 3.88 | 10.07 | 1.16 | 1.49 | 3.91 | 0.69 | 0.87 | **3.05** |
| cGCA (w/ cond.) | **1.75** | **2.33** | 5.38 | **0.87** | **1.08** | 2.96 | **0.57** | **0.64** | 2.34 |

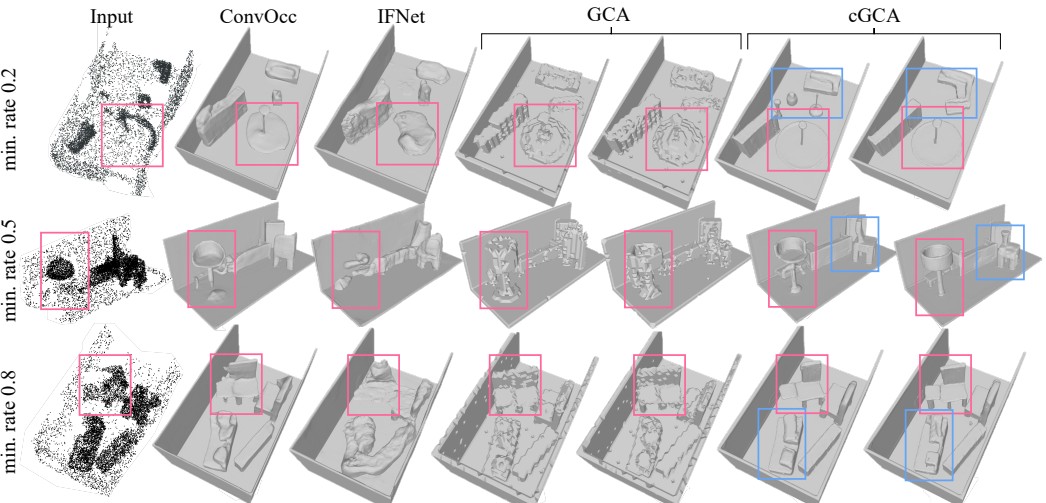

Figure 3: Qualitative comparison on ShapeNet scene dataset. Best viewed on screen. Minimum rate indicates the guaranteed rate of surface points for each object in the scene. While deterministic methods (ConvOcc, IFNet) produce blurry surfaces since they cannot model multi-modal distribution, probabilistic methods (GCA, cGCA) generate multiple plausible scenes. cGCA is the only method that can generate multiple plausible scenes without losing the details for each object.

## 4 EXPERIMENTS

We test the probabilistic shape completion of cGCA for scenes (Section 4.1) and single object (Section 4.2). For all the experiments, we use latent code dimension $K = 32$, trained with regularization parameter $\beta = 0.001$. All the results reported are re-trained for each dataset. Further implementation details are in the Appendix A.2 and effects of mode seeking steps are discussed in Appendix E.

### 4.1 SCENE COMPLETION

In this section, we evaluate our method on two datasets: ShapeNet scene (Peng et al. (2020)) and 3DFront (Fu et al. (2021)) dataset. The input point cloud are created by iteratively removing points within a fixed distance from a random surface point. We control the minimum preserved ratio of the original complete surface points for each object in the scene to test varying levels of completeness. The levels of completeness tested are 0.2, 0.5, and 0.8 with each dataset trained/tested separately. We evaluate the quality and diversity of the completion results by measuring the Chamfer-L1 distance (CD), total mutual distance (TMD), respectively, as in the previous methods (Peng et al. (2020); Zhang et al. (2021)). For probabilistic methods, five completion results are generated and we report the minimum and average of Chamfer-L1 distance (min. CD, avg. CD). Note that if the input is severely incomplete, there exist various modes of completion that might be feasible but deviate from its ground truth geometry. Nonetheless, we still compare CD assuming that plausible reconstructions are likely to be similar to the ground truth.

**ShapeNet Scene.** ShapeNet scene contains synthetic rooms that contain multiple ShapeNet (Chang et al. (2015)) objects, which have been randomly scaled with randomly scaled floor and random

Table 2: Quantitative comparison of probabilistic scene completion in 3DFront. The best results are marked as bold. Note that CD (quality, ↓) and TMD (diversity, ↑) in tables are multiplied by $10^3$.

| Method | min. rate 0.2 | | | min. rate 0.5 | | | min. rate 0.8 | | |
|---|---|---|---|---|---|---|---|---|---|
| | min. CD | avg. CD | TMD | min. CD | avg. CD | TMD | min. CD | avg. CD | TMD |
| GCA | 4.89 | 6.59 | **16.90** | 3.11 | 3.53 | **8.95** | 2.53 | 2.82 | **7.98** |
| cGCA | 4.07 | 5.42 | 16.20 | **2.12** | 2.57 | 7.82 | 1.64 | 2.19 | 5.97 |
| cGCA (w/ cond.) | **3.53** | **4.26** | 9.23 | 2.19 | **2.54** | 6.89 | **1.47** | **1.69** | 5.35 |

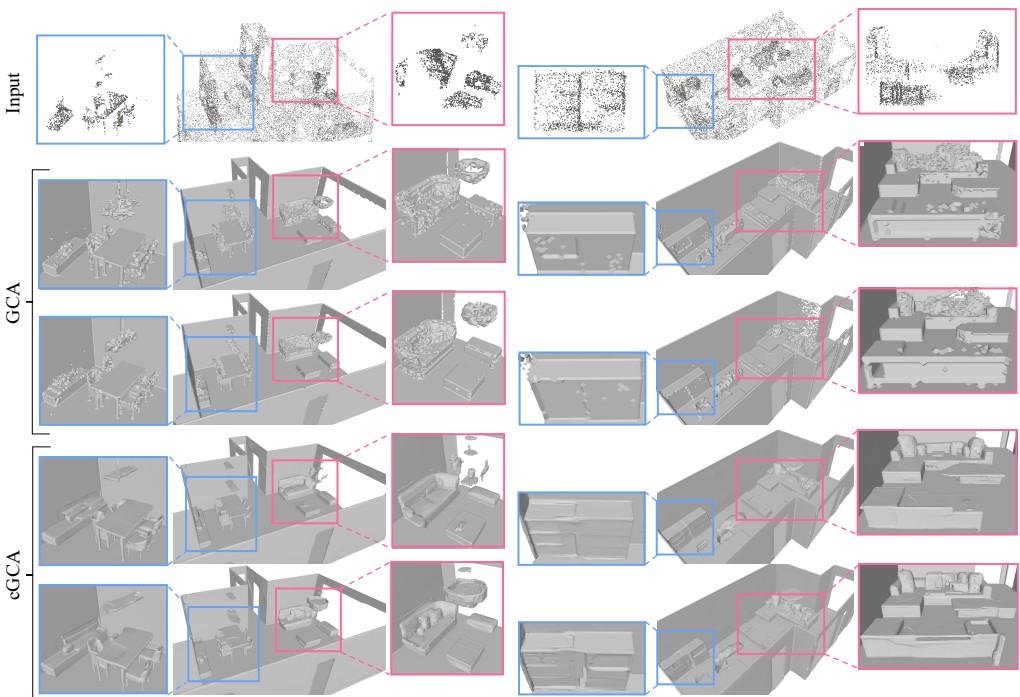

Figure 4: Qualitative comparison on 3DFront dataset with 0.5 object minimum rate. Best viewed on screen. Since the raw inputs of furniture are highly incomplete, there exist multiple plausible reconstructions. Probabilistic approaches produce diverse yet detailed scene geometry. GCA suffers from artifacts due to the discrete voxel resolution.

walls. We compare the performance of cGCA with two deterministic scene completion models that utilize occupancy as the implicit representation: ConvOcc (Peng et al. (2020)) and IFNet (Chibane et al. (2020a)). We additionally test the quality of our completion against a probabilistic method of GCA (Zhang et al. (2021)). Both GCA and cGCA use $64^3$ voxel resolution with $T = 15$ transitions. cGCA (w/ cond.) indicates a variant of our model, where each transition is conditioned on the initial state $s^0$ obtained by the encoder $g_\phi$ as discussed in Sec. 3.2.

Table 1 contains the quantitative comparison on ShapeNet scene. Both versions of cGCA outperform all other methods on min. CD for all level of completeness. The performance gap is especially prominent for highly incomplete input, which can be visually verified from Fig. 3. The deterministic models generate blurry objects given high uncertainty, while our method consistently generates detailed reconstructions for inputs with different levels of completeness. Our result coincides with the well-known phenomena of generative models, where the deterministic models fail to generate crisp outputs of multi-modal distribution (Goodfellow (2017)). Considering practical scenarios with irregular real-world scans, our probabilistic formulation is crucial for accurate 3D scene completion. When conditioned with the initial state, the completion results of cGCA stay faithful to the input data, achieving lower CDs. Further analysis on varying completeness of input is discussed in Appendix F.

**3DFront.** 3DFront is a large-scale indoor synthetic scene dataset with professionally designed layouts and contains high-quality 3D objects, in contrast to random object placement for ShapeNet scene. 3DFront dataset represents the realistic scenario where the objects are composed of multiple meshes without clear boundaries for inside or outside. Unless the input is carefully processed to be converted into a watertight mesh, the set-up excludes many of the common choices for implicit representation,

Table 3: Quantitative comparison of single object probabilistic shape completion results on ShapeNet. The best results trained in a single class are marked as bold. Note that MMD (quality, $\downarrow$), UHD (fidelity, $\downarrow$) and TMD (diversity, $\uparrow$) in tables are multiplied by $10^3$, $10^3$, and $10^2$ respectively.

| Method | MMD (quality) | | | | UHD (fidelity) | | | | TMD (diversity) | | | |
|---|---|---|---|---|---|---|---|---|---|---|---|---|
| | Sofa | Chair | Table | Avg. | Sofa | Chair | Table | Avg. | Sofa | Chair | Table | Avg. |
| cGAN | 5.70 | 6.53 | 6.10 | 6.11 | 11.40 | 12.10 | 11.20 | 11.57 | 7.44 | 8.21 | 6.88 | 7.51 |
| GCA ($64^3$) | 4.70 | 6.23 | 6.22 | 5.72 | 8.88 | 7.95 | 7.63 | 8.15 | **22.39** | 12.41 | 18.83 | 17.88 |
| cGCA ($32^3$) | 4.59 | **6.11** | 6.08 | 5.59 | 10.43 | 9.99 | 9.29 | 9.90 | 9.72 | 13.65 | 26.04 | 16.47 |
| cGCA ($64^3$) | **4.51** | 6.30 | **5.89** | 5.57 | 8.99 | 8.43 | 7.22 | 8.21 | 11.18 | **16.70** | **31.94** | **19.94** |
| cGCA ($64^3$, w/ cond.) | 4.64 | 6.15 | 5.90 | **5.56** | **8.00** | **6.75** | **6.45** | **7.07** | 14.01 | 11.49 | 31.01 | 18.84 |

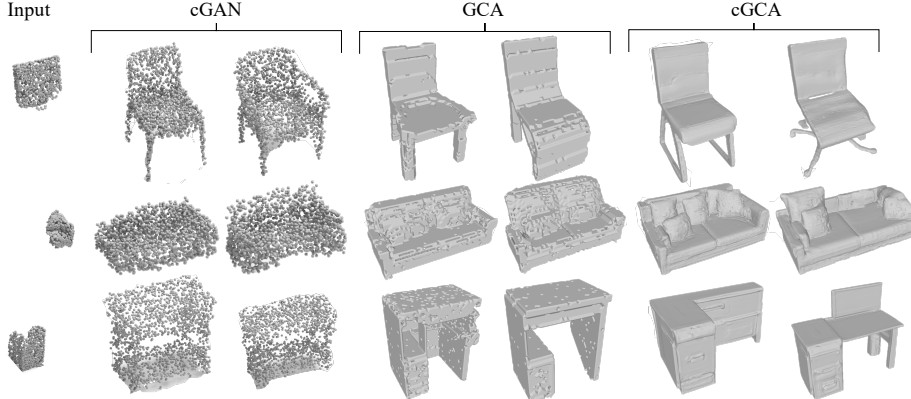

Figure 5: Qualitative comparison on probabilistic shape completion of a single object. cGCA is the only method that can produce a continuous surface.

such as occupancy or signed distance fields. However, the formulation of cGCA can be easily adapted for different implicit representation, and we employ *unsigned distance fields* (Chibane et al. (2020b)) to create the sparse voxel embedding for 3DFront dataset. We compare the performance of cGCA against GCA, both with voxel resolution of 5cm and $T = 20$ transitions.

Table 2 shows that cGCA outperforms GCA by a large margin in CD, generating high-fidelity completions with unsigned distance fields. While both GCA and cGCA are capable of generating multiple plausible results, GCA suffers from discretization artifacts due to voxelized representation, as shown in Fig. 4. cGCA not only overcomes the limitation of the resolution, but also is scalable to process the entire rooms at once during both training and test time. In contrast, previous methods for scene completion (Siddiqui et al.; Peng et al. (2020)) divide the scene into small sections and separately complete them. We analyze the scalability in terms of the network parameters and the GPU usage in Appendix D. In Appendix G, we also provide results on ScanNet (Dai et al. (2017a)) dataset, which is one of the widely used datasets for real-world indoor environments.

## 4.2 SINGLE OBJECT COMPLETION

We analyze various performance metrics of cGCA for a single object completion with chair/sofa/table classes of ShapeNet (Chang et al. (2015)) dataset. Given densely sampled points of a normalized object in $[-1, 1]^3$, the incomplete observation is generated by selecting points within the sphere of radius 0.5 centered at one of the surface points. From the partial observation, we sample 1,024 points which serve as a sparse input, and sample 2,048 points from completion results for testing. Following the previous method (Wu et al. (2020)), we generate ten completions and compare MMD (quality), UHD (fidelity), and TMD (diversity). Our method is compared against other probabilistic shape completion methods: cGAN (Wu et al. (2020)) which is based on point cloud, and GCA (Zhang et al. (2021)) which uses voxel representation. We use $T = 30$ transitions.

Quantitative and qualitative results are shown in Table 3 and Fig. 5. Our approach exceeds other baselines in all metrics, indicating that cGCA can generate high-quality completions (MMD) that are faithful to input (UHD) while being diverse (TMD). By using latent codes, the completed continuous surface of cGCA can capture geometry beyond its voxel resolution. The quality of completed shape in $32^3$ voxel resolution therefore even outperforms in MMD for discrete GCA in higher $64^3$ voxel

resolution. Also, the UHD score of cGCA (w/ cond.) exceeds that of GCA and vanilla cGCA indicating that conditioning latent codes from the input indeed preserves the input partial geometry.

## 5 RELATED WORKS

**3D Shape Completion.** The data-driven completion of 3D shapes demands a large amount of memory and computation. The memory requirement for voxel-based methods increases cubically to the resolution (Dai et al. (2017b)) while the counterpart of point cloud based representations (Yuan et al. (2018)) roughly increases linearly with the number of points. Extensions of scene completion in voxel space utilize hierarchical representation (Dai et al. (2018)) or subdivided scenes (Dai et al. (2018; 2020)) with sparse voxel representations. Recently, deep implicit representations (Park et al. (2019); Chen & Zhang (2019); Mescheder et al. (2019)) suggest a way to overcome the limitation of resolution. Subsequent works (Chabra et al. (2020); Chibane et al. (2020a); Jiang et al. (2020); Peng et al. (2020)) demonstrate methods to extend the representation to large-scale scenes. However, most works are limited to regressing a single surface from a given observation. Only a few recent works (Wu et al. (2020); Zhang et al. (2021); Smith & Meger (2017)) generate multiple plausible outcomes by modeling the distribution of surface conditioned on the observation. cGCA suggests a scalable solution for multi-modal continuous shape completion by employing progressive generation with continuous shape representations.

**Diffusion Probabilistic Models.** One way to capture the complex data distribution by the generative model is to use a diffusion process inspired by nonequilibrium thermodynamics such as Sohl-Dickstein et al. (2015); Ho et al. (2020); Luo & Hu (2021). The diffusion process incrementally destroys the data distribution by adding noise, whereas the transition kernel learns to revert the process that restores the data structure. The learned distribution is flexible and easy to sample from, but it is designed to evolve from a random distribution. On the other hand, the infusion training by Bordes et al. (2017) applies a similar technique but creates a forward chain instead of reverting the diffusion process. Since the infusion training can start from a structured input distribution, it is more suitable to a shape completion that starts from a partial data input. However, the infusion training approximates the lower bound of variational distribution with Monte Carlo estimates using the reprameterization trick (Kingma & Welling (2014)). We modify the training objective and introduce a simple variant of infusion training that can maximize the variational lower bound of the log-likelihood of the data distribution without using Monte Carlo approximation.

## 6 CONCLUSION

We are the first to tackle the challenging task of probabilistic scene completion, which requires not only the model to generate multiple plausible outcomes but also be scalable to capture the wide-range context of multiple objects. To this end, we propose continuous Generative Cellular Automata, a scalable generative model for completing multiple plausible continuous surfaces from an incomplete point cloud. cGCA compresses the implicit field into sparse voxels associated with their latent code named sparse voxel embedding, and incrementally generates diverse implicit surfaces. The training objective is proven to maximize the variational lower bound of the likelihood of sparse voxel embeddings, indicating that cGCA is a theoretically valid generative model. Extensive experiments show that our model is able to faithfully generate multiple plausible surfaces from partial observation.

There are a few interesting future directions. Our results are trained with synthetic scene datasets where the ground truth data is available. It would be interesting to see how well the data performs in real data with self-supervised learning. For example, we can extend our method to real scenes such as ScanNet Dai et al. (2017a) or Matterport 3D (Chang et al. (2017)) by training the infusion chain with data altered to have different levels of completeness as suggested by Dai et al. (2020). Also, our work requires two-stage training, where the transition kernel is trained with the ground truth latent codes generated from the pre-trained autoencoder. It would be less cumbersome if the training could be done in an end-to-end fashion. Lastly, our work takes a longer inference time compared to previous methods (Peng et al. (2020)) since a single completion requires multiple transitions. Reducing the number of transitions by using a technique similar to Salimans & Ho (2022) can accelerate the runtime.

## 7 ETHICS STATEMENT

The goal of our model is to generate diverse plausible scenes given an observation obtained by sensors. While recovering the detailed geometry of real scenes is crucial for many VR/AR and robotics applications, it might violate proprietary or individual privacy rights when abused. Generating the unseen part can also be regarded as creating fake information that can deceive people as real.

## 8 REPRODUCIBILITY

Code to run the experiments is available at https://github.com/96lives/gca. Appendix A contains the implementation details including the network architecture, hyperparameter settings, and dataset processing. Proofs and derivations are described in Appendix B.

## 9 ACKNOWLEDGEMENT

We thank Youngjae Lee for helpful discussion and advice on the derivations. This research was supported by the National Research Foundation of Korea (NRF) grant funded by the Korea government (MSIT) (No.2020R1C1C1008195) and the National Convergence Research of Scientific Challenges through the National Research Foundation of Korea (NRF) funded by Ministry of Science and ICT (NRF2020M3F7A1094300). Young Min Kim is the corresponding author.

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

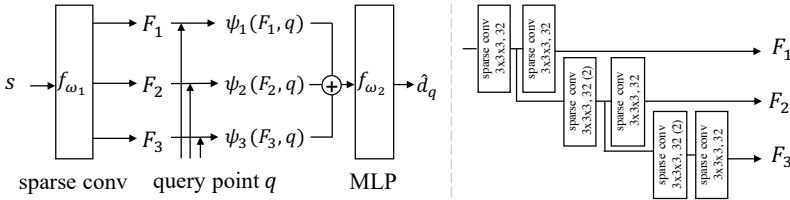

Figure 6: Neural network architecture for the decoder $f_\omega$. The left side shows the overall architecture for the decoder $f_\omega$ and the right side shows the architecture for sparse convolution layers $f_{\omega_1}$. The parenthesis denotes the stride of the sparse convolution and every convolution except the feature extracting layer is followed by batch normalization and ReLU activation.

# A  IMPLEMENTATION DETAILS

We use Pytorch (Paszke et al. (2019)) and the sparse convolution library MinkowskiEngine (Choy et al. (2019)) for all implementations.

## A.1  AUTOENCODER

**Network Architecture.** The autoencoder is composed of two modules, encoder $g_\phi$ and decoder $f_\omega$ to convert between the sparse voxel embedding and implicit fields, which is depicted in Fig. 2. The encoder $g_\phi$ generates the sparse voxel embedding $s$ from the coordinate-distance pairs $P = \{(p, d_p) | p \in \mathbb{R}^3, d_p \in \mathbb{R}\}$, where $p$ is a 3D coordinate and $d_p$ is the (signed) distance to the surface, $s = g_\phi(P)$. The decoder $f_\omega$ regresses the implicit field value $d_q$ at the 3D position $q \in \mathbb{R}^3$ given the sparse voxel embedding $s$, $d_q = f_\omega(q, s)$.

The encoder $g_\phi$ is a a local PointNet (Qi et al. (2016)) implemented with MLP of 4 blocks with hidden dimension 128, as in ConvOcc (Peng et al. (2020)). Given a set of coordinate-distance pair $P$, we find the nearest cell $c_p$ for each coordinate-distance pair $(p, d_p)$. Then we make the representation sparse by removing the coordinate-distance pair if the nearest cell $c_p$ do not contain the surface. For the remaining pairs, we normalize the coordinate $p$ to the local coordinate $p' \in [-1, 1]^3$ centered at the nearest voxel. Then we use the local coordinate to build a vector in $\mathbb{R}^4$ for coordinate-distance pair $(p', d_p)$ which serves as the input to the PointNet for the encoder $g_\phi$. The local PointNet architecture employs average pooling to the point-coordinate pairs that belong to the same cell.

The architecture of decoder $f_\omega$, inspired by IFNet (Chibane et al. (2020a)) is depicted in Fig. 6. As shown on the left, the decoder $f_\omega$ can be decomposed into the sparse convolution $f_{\omega_1}$ that extracts hierarchical feature, followed by trilinear interpolation of the extracted feature $\Psi$, and final MLP layers $f_{\omega_2}$. The sparse convolution layer $f_{\omega_1}$ aggregates the information into multi-level grids in $n$ different resolutions, i.e., $f_{\omega_1}(s) = F_1, F_2, ..., F_n$, Starting from the original resolution $F_1$, $F_k$ contains a grid that is downsampled $k$ times. However, in contrast to the original IFNet, our grid features are sparse, since they only store the downsampled occupied cells of sparse voxel embedding $s$. The sparse representation $F_k = \{(c, o_c, e_c) | c \in \mathbb{Z}^3, o_c \in \{0, 1\}, e_c \in \mathbb{R}^L\}$ is composed of occupancy $o_c$ and $L$-dimensional feature $e_c$ for each cell, like sparse voxel embedding. The grid points $c$ that are not occupied are considered as having zero features. We use $n = 3$ levels, with feature dimension $L = 32$ for all experiments.

The multi-scale features are then aggregated to define the feature for a query point $q$: $\Psi(f_{\omega_1}(s), q) = \sum_k \psi_k(F_k, q)$. Here $\psi_k(F_k, q) \in \mathbb{R}^L$ is the trilinear interpolation of features in discrete grid $F_k$ to define the feature at a particular position $q$, similar to IFNet. We apply trilinear interpolation at each resolution $k$ then combine the features.

Lastly, the MLP layer maps the feature at the query point $q$ to the implicit function value $\hat{d}_q$, $f_{\omega_2} : \mathbb{R}^L \to \mathbb{R}$. The MLP is composed of 4 ResNet blocks with hidden dimension 128 with $\tanh$ as the final activation function. The final value $\hat{d}_q$ is the truncated signed distance within the range of $[-1, 1]$ except for 3DFront (Fu et al. (2021)) which does not have a watertight mesh. We use unsigned distance function for 3DFront as in NDF (Chibane et al. (2020b)) by changing the codomain of the

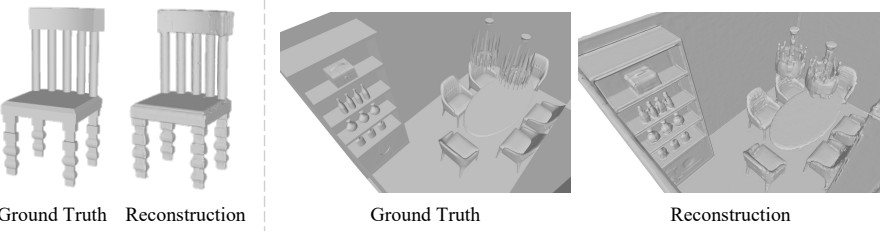

Ground Truth    Reconstruction        Ground Truth            Reconstruction

Figure 7: Visualizations of reconstructions by autoencoder. We visualize a reconstruction of a chair using signed distance fields (left) and a scene in 3DFront using unsigned distance fields (right). Both reconstructions show that sparse voxel embedding is able to reconstruct a continuous surface.

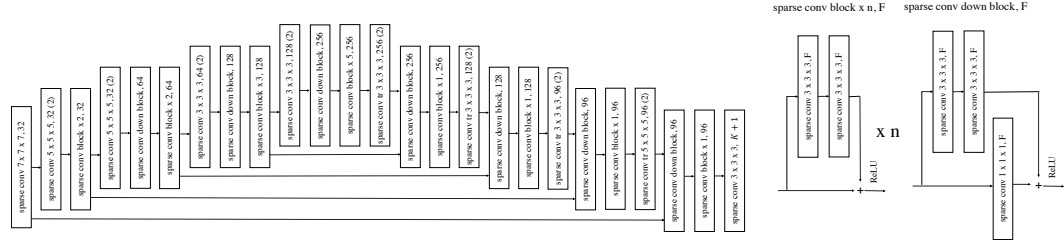

Figure 8: Neural network architecture for the transition model $p_\theta$. We employ the architecture of U-Net. The paranthesis indicates the stride of convolutions and each convolution is followed by batch normalization.

decoder to $[0, 1]$ with sigmoid as the final activation function. Thus, the decoder can be summarized as $f_\omega(s, q) = f_{\omega_2}(\Psi(f_{\omega_1}(s), q))$.

We would like to emphasize that the autoencoder does not use any dense 3D convolutions, and thus the implementation is efficient, especially when reconstructing the sparse voxel embedding. Since the decoding architecture only consists of sparse convolutions and MLPs, the memory requirement increases linearly with respect to the number of occupied voxels. In contrast, the requirement for dense 3D convolution increases cubic with respect to the maximum size of the scene.

**Reconstruction Results.** Fig. 7 shows a reconstruction of a chair (left) using signed distance fields and a scene in 3DFront (right) using unsigned distance fields. The figure shows that our autoencoder can faithfully reconstruct a continuous surface using the sparse voxel embedding. However, the usage of unsigned distance fields (Chibane et al. (2020b)) representation tends to generate relatively thicker reconstructions compared to that of signed distance fields. Without a clear distinction between the interior/exterior, the unsigned representation makes it difficult to scrutinize the zero-level isosurface and thus results in comparably thick reconstruction. Thus, the flexibility of being able to represent any mesh with various topology comes at a cost of expressive power, in contrast to signed distance fields which can only represent closed surfaces. Our work can be further improved by utilizing more powerful implicit representation techniques, such as SIREN (Sitzmann et al. (2020)), but we leave it as future work.

**Hyperparameters and Implementation Details.** We train the autoencoder with Adam (Kingma & Ba (2015)) optimizer with learning rate 5e-4 and use batch size of 3, 1, and 4 for ShapeNet scene (Peng et al. (2020)), 3DFront (Fu et al. (2021)), and ShapeNet (Chang et al. (2015)) dataset, respectively. The autoencoder for scene completion is pretrained with all classes of ShapeNet assuming that the local geometric details for scenes are represented with the patches from individual objects.

## A.2    TRANSITION MODEL

**Network Architecture.** Following GCA (Zhang et al. (2021)), we implement the transition model by using a variant of U-Net (Ronneberger et al. (2015)). The input to the sparse convolution network is the sparse voxel embedding $s^t$, and the output feature size of the last convolution is $K + 1$, where

we use generalized sparse convolution for the last layer to obtain the occupancy (with dimension 1) and the latent code (with dimension $K$) of the neighborhood. Unlike GCA, our implementation does not need an additional cell-wise aggregation step.

$s^0$ **Conditioned Model.** The $s^0$ conditioned model is a variant of our model, where we employ the use of encoder $g_\phi$ to estimate the initial latent codes of initial state $s^0$ and condition every transition kernel on $s^0$. Given incomplete point cloud $R = \{c_1, c_2, ..., c_M\}$ with $c_i \in \mathbb{R}^3$, the variant constructs the initial $s_0$ by encoding a set of coordinated-distance pair $P_R = \{(c, 0) | c \in R\}$, where the distance is set to 0 for every coordinate in $R$. Since the point cloud is sampled from the surface of the original geometry, this is a reasonable estimate of the latent code.

The transition kernel $p_\theta$ is conditioned on $s^0$ by taking a sparse voxel embedding $\hat{s}^t$ of latent code dimension $2K$ that concatenates the current state $s^t$ and $s^0$ as the input. Specifically, we define $\hat{s}^t$ to be $\hat{s}^t = \{(c, o_c, z_c) | o_c = o_c^{s^t} \vee o_c^{s^0}, z_c = [z_c^{s^0}, z_c^{s^t}] \in \mathbb{R}^{2K}, c \in \mathbb{Z}^3\}$, where the $o_c^{s^t}, o_c^{s^0}, z_c^{s^t}, z_c^{s^0}$ are the occupancies and latent codes of $s^t$ and $s^0$. The occupancy is set to one if either $s^t$ or $s^0$ is occupied and the latent code is set to be the concatenation of latent codes of $s^0$ and $s^t$. Thus, the only modification to the neural network $p_\theta$ is the size of input feature dimension, which is increased from $K$ to $2K$.

**Hyperparameters and Implementation Details** We train our model with Adam (Kingma & Ba (2015)) optimizer with learning rate 5e-4. The transition kernel is trained using infusion schedule $\alpha^t = 0.1 + 0.005t$ and standard deviation schedule $\sigma^t = 10^{-1-0.01t}$. We use the accelerated training scheme introduced in GCA (Zhang et al. (2021)) by managing the time steps based on the current step. We use neighborhood size $r = 2$, with metric $d$ induced by $L_1$ norm, except for the ShapeNet (Chang et al. (2015)) experiment on $64^3$ voxel resolution which uses $r = 3$. For transition model training, we utilize maximum batch size of 32, 16 and 6, for ShapeNet scene, 3DFront, ShapeNet dataset, respectively. We use adaptive batch size for the transition model, which is determined by the total number of cells in the minibatch for efficient GPU memory usage. Lastly, we use mode seeking steps $T' = 5$ for all experiments.

**Visualizations and Testing.** All surface visualizations are made by upsampling the original voxel resolution 4 times, where we efficiently query from the upsampled grid only near the occupied cells of the final state $s^{T+T'}$. Then, the mesh is extracted with marching cubes (Lorensen & Cline (1987)) with no other postprocessing applied. From the $4\times$ upsampled high resolution grid, we extract cells with the distance value below 0.5 which serve as the point cloud to compute the metrics for testing.

## A.3 BASELINES

We use the official implementation or the code provided by the authors by contact with default hyperparameters unless stated otherwise. All results are obtained by training on our dataset. The output point cloud used for testing, is obtained by sampling from mesh with the same resolution if possible, unless stated otherwise. For GCA (Zhang et al. (2021)) we use the same hyperparameters as ours for fair comparison. The mesh is created by using marching cubes (Lorensen & Cline (1987)) with the voxel resolution. We use the volume $64^3$ encoder for ConvOcc (Peng et al. (2020)) which achieves the best results in the original ShapeNet scene dataset (Peng et al. (2020)). For IFNet (Chibane et al. (2020a)), we use the ShapeNet128Vox model as used in the work of Siddiqui et al. with occupancy implicit representation. cGAN (Wu et al. (2020)) is tested by 2,048 generated point cloud.

## A.4 DATASETS

**ShapeNet Scene.** ShapeNet scene (Peng et al. (2020)) contains synthetic rooms that contain multiple ShapeNet (Chang et al. (2015)) objects, which has randomly scaled floor and random walls, normalized so that the longest length of the bounding box has length of 1. The number of rooms for train/val/test split are 3750/250/995. We sample SDF points for training our model. The input is created by iteratively sampling points from the surface and removing points within distance 0.1 for each instance. We run 20 iterations and each removal is revoked if the aggregated removal exceeds that of the guaranteed rate. Since we are more interested in the reconstruction of objects, the walls and floors are guaranteed to have minimum rate of 0.8, regardless of the defined minimum rate. Lastly,

we sample 10,000 points and add normal noise with standard deviation 0.005. For computing the metrics at test time, we sample 100,000 points.

**3DFront.** 3DFront (Fu et al. (2021)) is a large-scale indoor synthetic scene dataset with professionally designed layouts and contains high quality 3D objects. We collect rooms that contain multiple objects, where 10% of the biggest rooms are removed from the data to discard rooms with immense size for stable training. Thus, 7044/904/887 rooms are collected as train/val/test rooms. The incomplete point cloud is obtained by randomly sampling with density 219 points/m$^2$ and then removed and noised as in ShapeNet scene, but with removal sphere distance proportional to the object size and noise of standard deviation of 1cm. During evaluation, we sample points proportionally to the area for each ground truth surface with density of 875 points/m$^2$ to compute the metrics.

**ShapeNet.** We use chair, sofa, table classes of ShapeNet (Chang et al. (2015)), where the signed distance values of the query points are obtained by using the code of DeepSDF (Park et al. (2019)). The shapes are normalized to fit in a bounding box of $[-1, 1]^3$. The partial observation is generated by extracting points within a sphere of radius 0.5, centered at one of the surface points. Then we sample 1,024 points to create a sparse input and compare the metrics by sampling 2,048 points for a fair comparison against cGAN (Wu et al. (2020)).

## B  PROOFS AND DETAILED DERIVATIONS

### B.1  VARIATIONAL BOUND DERIVATION

We derive the variational lower bound for cGCA (Eq. (11)) below. The lower bound is derived by the work of Sohl-Dickstein et al. (2015) and we include it for completeness.

$$
\begin{aligned}
\log p_\theta(x) &= \log \sum_{s^{0:T-1}} p_\theta(s^{0:T-1}, x) \\
&= \log \sum_{s^{0:T-1}} q_\theta(s^{0:T-1}|x) \frac{p_\theta(s^{0:T-1}, x)}{q_\theta(s^{0:T-1}|x)} \\
&\geq \sum_{s^{0:T-1}} q_\theta(s^{0:T-1}|x) \log \frac{p_\theta(s^{0:T-1}, x)}{q_\theta(s^{0:T-1}|x)} \quad (\because \text{Jensen's inequality}) \\
&= \sum_{s^{0:T-1}} q_\theta(s^{0:T-1}|x)(\log \frac{p(s^0)}{q(s^0)} + \sum_{0 \leq t < T-1} \log \frac{p_\theta(s^{t+1}|s^t)}{q_\theta(s^{t+1}|s^t, x)} + \log p(x|s^{T-1})) \\
&= \log \frac{p(s^0)}{q(s^0)} + \sum_{s^{0:T-1}} q_\theta(s^{0:T-1}|x) \sum_{0 \leq t < T-1} \log \frac{p_\theta(s^{t+1}|s^t)}{q_\theta(s^{t+1}|s^t, x)} + \mathbb{E}_{q_\theta}[\log p_\theta(x|s^{T-1})]
\end{aligned}
$$

The second term of the right hand side can be converted into KL divergence as following:

$$
\begin{aligned}
&\sum_{s^{0:T-1}} q_\theta(s^{0:T-1}|x) \sum_{0 \leq t < T-1} \log \frac{p_\theta(s^{t+1}|s^t)}{q_\theta(s^{t+1}|s^t, x)} \\
&= \sum_{s^{0:T-1}} \prod_{0 \leq i < T-1} q_\theta(s^{i+1}|s^i, x) \sum_{0 \leq t < T-1} \log \frac{p_\theta(s^{t+1}|s^t)}{q_\theta(s^{t+1}|s^t, x)} \\
&= \sum_{s^{0:T-1}} \sum_{0 \leq t < T-1} \prod_{\substack{0 \leq i < T-1 \\ i \neq t}} q_\theta(s^{i+1}|s^i, x) q_\theta(s^{t+1}|s^t, s^T) \log \frac{p_\theta(s^{t+1}|s^t)}{q_\theta(s^{t+1}|s^t, x)} \\
&= \sum_{0 \leq t < T-1} \sum_{s^{t+1}} ( \sum_{s^{-(t+1)}} \prod_{\substack{0 \leq i < T-1 \\ i \neq t}} q_\theta(s^{i+1}|s^i, x)) q_\theta(s^{t+1}|s^t, x) \log \frac{p_\theta(s^{t+1}|s^t)}{q_\theta(s^{t+1}|s^t, x)} \\
&\quad (s^{-(t+1)} \text{ denotes variables } s^{0:T-1} \text{ except } s^{t+1}) \\
&= \sum_{0 \leq t < T-1} \sum_{s^{t+1}} q_\theta(s^{t+1}|s^t, x) \log \frac{p_\theta(s^{t+1}|s^t)}{q_\theta(s^{t+1}|s^t, x)} \\
&= \sum_{0 \leq t < T-1} -\sum_{s^{t+1}} q_\theta(s^{t+1}|s^t, x) \log \frac{q_\theta(s^{t+1}|s^t, x)}{p_\theta(s^{t+1}|s^t)} \\
&= \sum_{0 \leq t < T-1} -D_{KL}(q_\theta(s^{t+1}|s^t, x) \| p_\theta(s^{t+1}|s^t))
\end{aligned}
$$

Thus,

$$
\log p_\theta(x) \geq \log \frac{p(s^0)}{q(s^0)} + \sum_{0 \leq t < T-1} -D_{KL}(q_\theta(s^{t+1}|s^t, x) \| p_\theta(s^{t+1}|s^t))) + \mathbb{E}_{q_\theta}[\log p_\theta(x|s^{T-1})]
$$

is derived.

## B.2   KL DIVERGENCE DECOMPOSITION

We derive the decomposition of KL divergence (Eq. (12) ) below.

$$
\begin{aligned}
\mathcal{L}_t &= D_{KL}(q_\theta(s^{t+1}|s^t, x)\|p_\theta(s^{t+1}|s^t)) \\
&= \sum_{c\in\mathcal{N}(s^t)} D_{KL}(q_\theta(o_c, z_c|s^t, x)\|p_\theta(o_c, z_c|s^t)) \quad (\because \text{Eq. (4), Eq.( 8)}) \\
&= \sum_{c\in\mathcal{N}(s^t)} D_{KL}(q_\theta(o_c|s^t, x)\|p_\theta(o_c|s^t)) \\
&\quad + \sum_{o_c\in\{0,1\}} q_\theta(o_c|s^t, x)D_{KL}(q_\theta(z_c|s^t, x, o_c)\|p_\theta(z_c|s^t, o_c)) \\
&\quad (\because \text{Eq. (4), Eq. (8) and chain rule for conditional KL divergence}) \\
&= \sum_{c\in\mathcal{N}(s^t)} D_{KL}(q_\theta(o_c|s^t, x)\|p_\theta(o_c|s^t)) \\
&\quad + q_\theta(o_c = 1|s^t, x)D_{KL}(q_\theta(z_c|s^t, x, o_c = 1)\|p_\theta(z_c|s^t, o_c = 1)) \\
&\quad (\because p_\theta(z_c|s^t, o_c = 0) = q_\theta(z_c|s^t, x, o_c = 0) = \delta_0(z_c))
\end{aligned}
$$

Thus, Eq. (12) is derived.

## B.3   APPROXIMATING $\mathcal{L}_{\text{final}}$ BY $\mathcal{L}_{T-1}$

In this section, we claim that $\mathcal{L}_{\text{final}}$ can be replaced by $\mathcal{L}_{T-1}$, which allows us to train using a simplified objective. First, we show that $\mathcal{L}_{\text{final}} = \mathbb{E}_{q_\theta}[\log p_\theta(x|s^{T-1})] = \infty$ due to the usage of Dirac distribution and introduce a non-diverging likelihood by replacing the Dirac function $\delta_0(z_c)$ with indicator function $\mathbb{1}[z_c = 0]$. Recall,

$$
\begin{aligned}
\mathcal{L}_{\text{final}} &= \mathbb{E}_{q_\theta}[\log p_\theta(x|s^{T-1})] \\
&= \sum_{c\in\mathcal{N}(s^{T-1})} \log\{(1-\lambda_{\theta,c})\mathbb{1}[o_c^x = 0]\delta_0(z_c^x) + \lambda_{\theta,c}\mathbb{1}[o_c^x = 1]N(z_c^x;\ \mu_{\theta,c}, \sigma^{T-1}\boldsymbol{I})\}
\end{aligned}
$$

Thus, if any cell $c \in \mathcal{N}(s^{T-1})$ is unoccupied, the likelihood diverges to $\infty$ since $z_c^x = 0$. This disables us to use negative log-likelihood as a loss function, since all the gradients will diverge.

This problem can be easily resolved by substituting $\delta_0(z_c^x)$ with indicator function $\mathbb{1}[z_c^x = 0]$. Both functions have non-zero value only at $z_c \neq 0$, but the former has value of $\infty$ while the latter has value 1. While $\mathbb{1}[z_c^x = 0]$ is not a probability measure, i.e. $\int_{z_c^x} \mathbb{1}[z_c^x = 0] \neq 1$, replacing $\delta_0(z_c^x)$ will have the effect of reweighting the likelihood at $z_c^x = 0$ from $\infty$ to 1, with the likelihood at other values $z_c^x \neq 0$ unchanged. Thus, $\mathbb{1}[z_c^x = 0]$ is a natural replacement of $\delta_0(z_c^x)$ for computing a valid likelihood.

**Proposition 1.** *By replacing $\delta_0(z_c^x)$ with the indicator function $\mathbb{1}[z_c^x = 0]$ when computing $\mathcal{L}_{\text{final}}$, $\nabla\mathcal{L}_{T-1} = \nabla\mathcal{L}_{\text{final}}$, for $T \gg 1$*

*Proof.* For brevity of notations, we define

$$\sigma = \sigma^t,$$

$$\lambda_{q,c} = (1-\alpha^t)\lambda_{\theta,c} + \alpha^t\mathbb{1}[c \in G_x(s^t)],$$

$$\mu_{q,c} = (1-\alpha^t)\mu_{\theta,c} + \alpha^t z_c^x,$$

where $\lambda_{q,c}, \mu_{q,c}$ is the occupancy probability, and mean of the infusion kernel $q$ at cell $c$. With $T \gg 1$, there exists $T_1 < T$, such that $\alpha^{T_1} = 1$, since $\alpha^t = \max(\alpha_1 t + \alpha_0, 1)$. Since $d(c, c') = 0$ if and only if $c = c'$, $G_x(s^t) = \{\text{argmin}_{c\in\mathcal{N}(s^t)}d(c, c') \mid c' \in x\}$ must converge to a set of occupied coordinates of $x$ since the distance decreases for each cell $c \in x$ at every step. This indicates that $\lambda_{q,c} = \mathbb{1}[c \in G_x(s^{T-1})] = o_c^x$. Also, $\mu_{q,c} = z_c^x$ holds for all $t > T_1$.

Then, $\mathcal{L}_{T-1}$ and $\mathcal{L}_{\text{final}}$ can be expressed as the following:

$$
\begin{aligned}
\mathcal{L}_{T-1} &= -D_{KL}(q_\theta(s^T|s^{T-1}, x)\|p_\theta(s^T|s^{T-1})) \\
&= -\sum_{c\in\mathcal{N}(s^{T-1})} D_{KL}(q_\theta(o_c^T|s^{T-1}, x)\|p_\theta(o_c^T|s^{T-1})) \\
&\quad + q_\theta(o_c^T = 1|s^{T-1}, x)D_{KL}(q_\theta(z_c|s^{T-1}, x, o_c^T = 1)\|p_\theta(z_c|s^{T-1}, o_c^T = 1)) \quad (\because \text{Eq. (12)}) \\
&= -\sum_{c\in\mathcal{N}(s^{T-1})} \lambda_{q,c}\log\frac{\lambda_{q,c}}{\lambda_{\theta,c}} + (1 - \lambda_{q,c})\log\frac{1 - \lambda_{q,c}}{1 - \lambda_{\theta,c}} + \lambda_{q,c}\frac{1}{2\sigma^2}\|\mu_{\theta,c} - \mu_{q,c}\|^2 \\
&= -\sum_{c\in\mathcal{N}(s^{T-1})} o_c^x\log\frac{o_c^x}{\lambda_{\theta,c}} + (1 - o_c^x)\log\frac{1 - o_c^x}{1 - \lambda_{\theta,c}} + o_c^x\frac{1}{2\sigma^2}\|z_c^x - \mu_{q,c}\|^2 \\
&\quad (\because \lambda_{\theta,c} = o_c^x, \mu_{q,c} = z_c^x)
\end{aligned}
$$

$$
\begin{aligned}
\mathcal{L}_{\text{final}} &= \mathbb{E}_{q_\theta}[\log p_\theta(x|s^{T-1})] \\
&= \sum_{c\in\mathcal{N}(s^{T-1})} \log\{(1 - \lambda_{\theta,c})\mathbb{1}[o_c^x = 0]\delta_0(z_c^x) + \lambda_{\theta,c}\mathbb{1}[o_c^x = 1]N(z_c^x; \mu_{\theta,c}, \sigma^{T-1}\mathbf{I})\} \\
&= \sum_{c\in\mathcal{N}(s^{T-1})} \log\{(1 - \lambda_{\theta,c})\mathbb{1}[o_c^x = 0]\mathbb{1}[z_c^x = 0] \\
&\quad + \lambda_{\theta,c}\mathbb{1}[o_c^x = 1]\exp(-\frac{1}{2\sigma^2}\|z_c^x - \mu_{\theta,c}\|_2^2)(2\pi\sigma)^{-\frac{K}{2}}\}
\end{aligned}
$$

We divide cases for $o_c^x = 0$ and $o_c^x = 1$. When $o_c^x = 0$,

$$
\mathcal{L}_{T-1} = \mathcal{L}_{\text{final}} = \sum_{c\in\mathcal{N}(s^{T-1})} \log(1 - \lambda_{\theta,c}),
$$

where we set $y\log y = 0$ for $y = 0$, since $\lim_{y\to 0} y\log y = 0$. Analogously, when $o_c^x = 1$,

$$
\mathcal{L}_{T-1} = \sum_{c\in\mathcal{N}(s^{T-1})} \log\lambda_{\theta,c} - \frac{1}{2\sigma^2}\|z_c^x - \mu_{\theta,c}\|^2
$$

$$
\mathcal{L}_{\text{final}} = \sum_{c\in\mathcal{N}(s^{T-1})} \log\lambda_{\theta,c} - \frac{1}{2\sigma^2}\|z_c^x - \mu_{\theta,c}\|^2 - \frac{K}{2}\log(2\pi\sigma).
$$

Thus, if $o_c^x = 1$, $\mathcal{L}_{T-1} = \mathcal{L}_{\text{final}} - \frac{K}{2}\log(2\pi\sigma)$, where $-\frac{K}{2}\log(2\pi\sigma)$ is a constant. We can conclude that $\nabla_\theta\mathcal{L}_{T-1} = \nabla_\theta\mathcal{L}_{\text{final}}$ for all $o_c^x$.

$\square$

Table 4: Number of neural network parameters and GPU memory usage comparison for different grid size with 3DFront dataset. The grid size indicates the largest length of the scene with voxel resolution 5cm and the unit of GPU memory is GB.

| Method | # of parameters (x$10^7$) | grid size | | | | |
| | | 55 | 77 | 93 | 124 | 235 |
|---|---|---|---|---|---|---|
| ConvOcc | 0.42 | 1.32 | 1.81 | 2.38 | 4.13 | 21.68 |
| cGCA | 4.12 | 2.40 | 2.81 | 2.88 | 2.84 | 3.24 |

## C    DIFFERENCE OF FORMULATION COMPARED TO GCA

We elaborate the difference of formulations compared to GCA (Zhang et al. (2021)).

**Usage of Latent Code.** The biggest difference between GCA and our model is the usage of local latent codes (Jiang et al. (2020); Chabra et al. (2020)) to generate continuous surface. Although the formulation of GCA allows scalable sampling of high resolution voxel, it cannot produce continuous surface due to predefined voxel resolution. However, we define a new state named sparse voxel embedding by appending local latent codes to the voxel occupancy. Sparse voxel embedding can be decoded to produce continuous geometry, outperforming GCA in accuracy in all the evaluated datasets in Sec. 4.

**Training Disconnected Objects.** During training, GCA assumes a property named partial connectivity between an intermediate state $s^t$ and ground truth state $x$ to guarantee the convergence of infusion sequences to ground truth $x$ (Property 1 and Proposition 1 of Zhang et al. (2021)). The connectivity means that any cell in $s^t$ can reach any cell in $x$ following the path of cells in $x$ with local transitions bounded by neighborhood size $r$. The assumption is required since the infusion kernel of single cell $q^t(o_c|s^t)$ for GCA, defined as

$$q^t(o_c|s^t) = Ber((1 - \alpha^t)\lambda_{\theta,c} + \alpha^t \mathbb{1}[c \in x]), \qquad (14)$$

inserts the ground truth cell $c \in x$ if the cell is within the neighborhood of current state $\mathcal{N}(s^t)$, i.e. $c \in \mathcal{N}(s^t) \cap x$. However, we eliminate the assumption by defining an auxiliary function

$$G_x(s) = \{\arg\min_{c \in \mathcal{N}(s)} d(c, c')|c' \in x\}, \qquad (15)$$

and modifying the formulation as

$$q^t(o_c|s^t) = Ber((1 - \alpha^t)\lambda_{\theta,c} + \alpha^t \mathbb{1}[c \in G_x(s^t)]), \qquad (16)$$

by switching $x$ in the infusion term to $G_x(s^t)$ from the original infusion kernel. The usage of $G_x(s^t)$ guarantees infusing a temporary ground truth cell $c$ that assures reaching all cells in $x$ regardless of the connectivity. The assumption is no longer required and therefore our training procedure is a general version of GCA.

**Training Procedure for Maximizing Log-likelihood.** Both GCA and our model utilizes the infusion training (Bordes et al. (2017)), by emulating the sequence of states that we aim to learn. The objective of GCA does not explicitly show the relationship between the training objective and maximizing the log-likelihood, which is what we aim to achieve. In Sec. 3.3, we give a detailed explanation of how our training objective approximates maximizing the lower bound of log-likelihood of data distribution. Thus, our training procedure completes the heuristics used in original work of GCA.

## D    ANALYSIS ON SCALABILITY

In this section, we investigate the scalability of our approach. Scene completion needs to process the 3D geometry of large-scale scenes with multiple plausible outputs, and sparse representation is crucial to find the diverse yet detailed solutions.

As the scale of the 3D geometry increases, the approaches utilizing the dense convolutional network cannot observe the entire scene at the same time. As a simple remedy, previous methods employ sliding window techniques for 3D scene completion (Siddiqui et al.; Peng et al. (2020)). Basically they divide the input scene into small segments, complete each segment, and then fuse them. The

formulation can only observe the information that is within the same segment for completion, and assumes that the divisions contain all the information necessary to complete the geometry. The size of the maximum 3D volume that recent works utilize amounts to a 3D box whose longest length is 3.2m for $64^3$ resolution voxel, where individual cells are about 5 cm in each dimension. Note that the size is comparable to ordinary furniture, such as sofa or dining table, whose spatial context might not be contained within the same segment. The relative positions between relevant objects can be ignored, and the missing portion of the same object can accidentally be separated into different segments. With our sparse representation, the receptive field is large enough to observe the entire room, and we can stably complete the challenging 3D scenes with multiple objects capturing the mutual context.

We further provide the numerical comparison of our sparse representation against the dense convolution of ConvOcc (Peng et al. (2020)). Table 4 shows the number of neural network parameters and the GPU usage for processing single rooms of different sizes in 3DFront (Fu et al. (2021)). We used 5 cm voxel resolution and the grid size indicates the number of cells required to cover the longest length of the bounding box for the room. The maximum GPU usage during 25 transitions of cGCA is provided. We did not include the memory used for the final reconstruction which regresses the actual distance values at dense query points, since the memory usage differs depending on the number of query points.

Our neural network is powerful, employing about 10 times more parameters than ConvOcc. With the efficient sparse representation, we can design the network to capture larger receptive fields with deeper network, enjoying more expressive power. This is reflected in the detailed reconstruction results of our approach. The memory usage is similar for smaller rooms for both approaches, but cGCA becomes much more efficient for larger rooms. When the grid size for room reaches 235, our method consumes 7 times less GPU memory compared to ConvOcc. As widely known, the memory and computation for 3D convolution increases cubic to the grid resolution with dense representation, while those for sparse representations increase with respect to the number of occupied voxels, which is much more efficient. Therefore our sparse representation is scalable and crucial for scene completion.

## E  EFFECTS OF MODE SEEKING STEPS

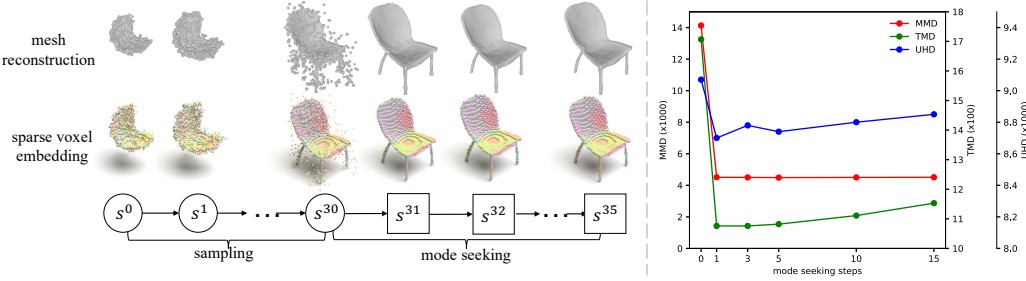

Figure 9: Ablation study on the effects of mode seeking step $T'$. The left shows an example of chair generation, where the bottom row are sparse voxel embeddings and the top row is the mesh reconstruction of each state. The right is the result on the effects of mode seeking step $T'$ tested with probabilistic shape completion on sofa dataset.

We perform an ablation study on the effects of mode seeking steps, which removes the noisy voxels with low-occupancy probability as discussed in Sec. 3.2. We test with ShapeNet dataset (Chang et al. (2015)) with voxel size $64^3$ and neighborhood radius $r = 3$, which tends to show the most intense effect of mode seeking phase due to large neighborhood size.

The right side of Fig. 9 shows the graph of metrics on quality (MMD), fidelity (UHD), and diversity (TMD) with respect to mode seeking steps. There is a dramatic decrease in all metrics on the first step of mode seeking step. The effect is visualized in the left side of Fig. 9, where the unlikely voxels are removed in a single mode seeking step. After the first mode seeking step, UHD and TMD metrics increase slightly, but overall, the metrics remain stable. We choose $T' = 5$ for all the conducted experiments to ensure that shape reaches a stable mode.

Table 5: Quantitative comparison of probabilistic scene completion in ShapeNet scene dataset with different levels of sparsity. The best results are marked as bold. Both CD (quality, $\downarrow$) and TMD (diversity, $\uparrow$) in tables are multiplied by $10^4$.

| Method | 500 points | | | 1,000 points | | | 5,000 points | | | 10,000 points | | |
| | min. CD | avg. CD | TMD | min. CD | avg. CD | TMD | min. CD | avg. CD | TMD | min. CD | avg. CD | TMD |
|---|---|---|---|---|---|---|---|---|---|---|---|---|
| ConvOcc | 36.58 | - | - | 5.92 | - | - | 1.65 | - | - | 1.33 | - | - |
| IFNet | 15.27 | - | - | 12.33 | - | - | 9.64 | - | - | 8.55 | - | - |
| GCA | 6.41 | 9.77 | 31.83 | 4.08 | 5.90 | **16.58** | 3.08 | 3.68 | **5.64** | 3.02 | 3.54 | **4.92** |
| cGCA | 11.30 | 17.61 | **50.58** | 3.64 | 5.53 | 16.18 | 1.32 | 1.73 | 4.64 | 1.16 | 1.49 | 3.91 |
| cGCA (w/ cond.) | **4.62** | **5.93** | 10.71 | **2.11** | **2.76** | 5.75 | **0.97** | **1.27** | 3.41 | **0.87** | **1.08** | 2.96 |

## F ANALYSIS ON VARYING COMPLETENESS OF INPUT

In this section, we further investigate the effects on the level of completeness of input. Sec. F.1 shows results of completion models on sparse inputs and Sec. F.2 provides how our model behaves when a non-ambiguous input is given.

### F.1 RESULTS ON SPARSE INPUT

For the experiments of ShapeNet scene dataset, the models are trained and evaluated on the input where the ambiguity lies on the missing parts, but the remaining parts of input pointcloud are relatively clear by sampling 10,000 points, following the experiment setting of ConvOcc (Peng et al. (2020)). In this section, we further provide ambiguity to the remaining parts by sampling less points. For the models trained on 10,000 point samples with 0.5 completeness, we evaluate the results on sparse input by sampling 500, 1,000, 5,000 and 10,000 points after applying the same iterative removal process as done in training. As in Sec. 4.1, we measure the accuracy (CD) and diversity (TMD) metrics compared with ConvOcc (Peng et al. (2020)), IFNet (Chibane et al. (2020a)), and GCA (Zhang et al. (2021)).

Table 5 contains the quantitative comparison results. For all the models, accuracy (CD) degrades as the input gets sparser. However, cGCA (w/ cond.) achieves best accuracy at all times, indicating that our model is the most robust to the sparsity of input. For probabilistic models, the diversity (TMD) increases as the density decreases. This implies that the diversity of the reconstructions is dependent to the ambiguity of the input. This is intuitively a natural response, since the multi-modality of plausible outcomes increases as the partial observations become less informative.

Fig. 10 visualizes the results for various sparsity. All of the models produce less plausible reconstructions with 500 points (leftmost column of Fig. 10) compared to that of higher density inputs. However, we observe a clear distinction between the completions of probabilistic models (GCA, cGCA) and deterministic models (ConvOcc, IFNet). While probabilistic models try to generate the learned shapes from the observations, such as a shade of the lamp, deterministic methods tend to fill the gaps between points. As discussed in Sec. 4.1, we hypothesize that this consequence stems from the fact that deterministic methods produce blurry results since it is forced to generate a single result along out of multiple plausible shapes which are the modes of the multi-modal distribution (Goodfellow (2017)). Therefore, we claim that a generative model capable of modeling multi-modal distribution should be used for shape completion task.

### F.2 RESULTS ON NON-AMBIGUOUS INPUT

We present experiments where cGCA is given a non-ambiguous input. For the models trained on various level of completeness on ShapeNet scene dataset, we evaluate each model on an input presenting distinct geometry. The input is generated by sampling 100,000 points from the mesh without any procedure of removal.

Table 6 shows the quantitative results of the experiments. The first row shows the accuracy (CD) and diversity (TMD) scores of the models operated on test set when the removal procedure is same as that of the training set. The second row shows the metrics when the input of the test set is sampled densely (100,000 points) without any removal. For all models trained with varying level of completeness, accuracy increases with decreasing diversity given a clear shape compared to an ambiguous shape. The result indicates that the diversity of the trained models is dependent on the ambiguity of the input.

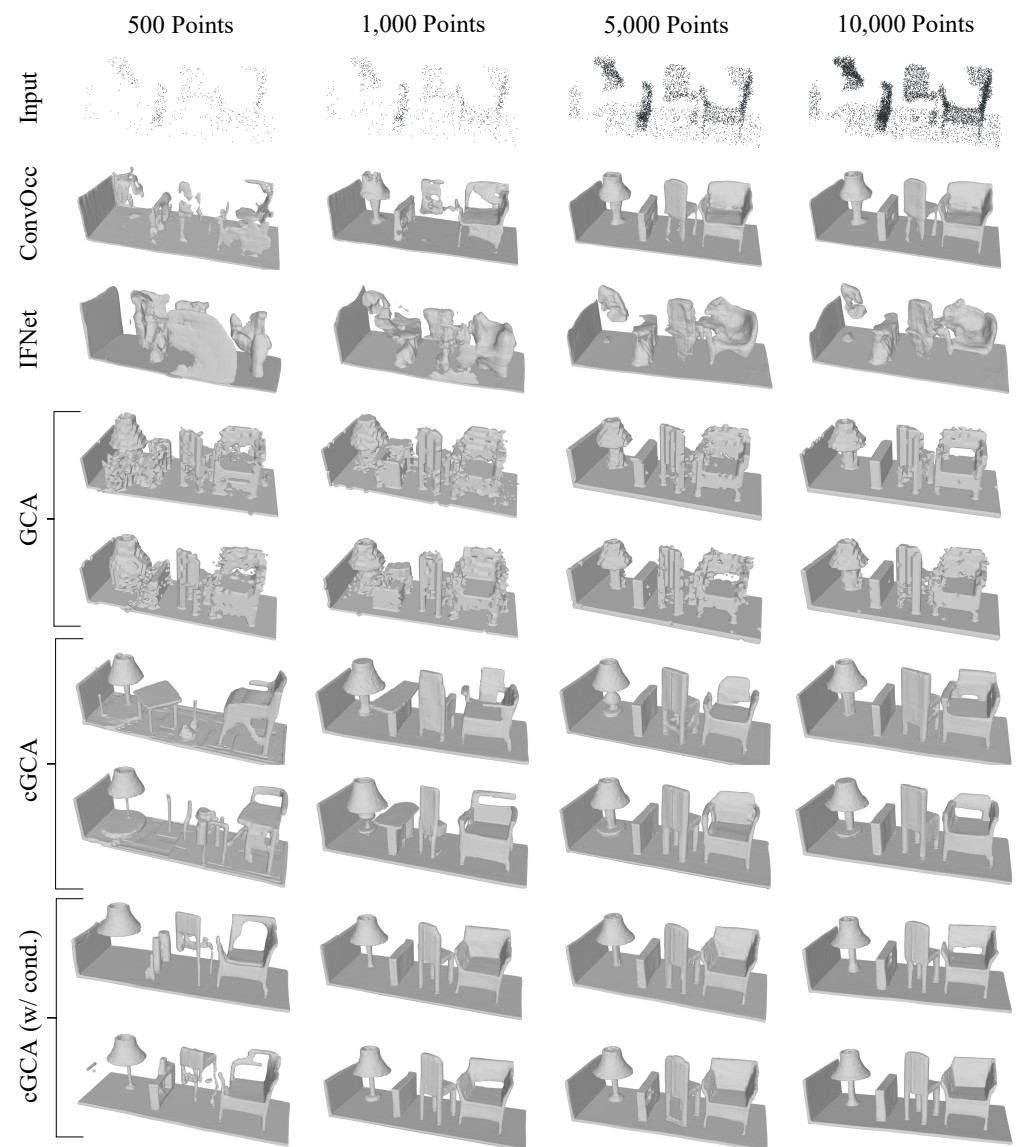

Figure 10: Qualitative results on ShapeNet scenes with varying level of density. Note that all the models are trained on dataset containing 10,000 points (rightmost column), but only tested with different density. While the probabilistic methods (GCA, cGCA) tries to generate the learned shapes (e.g. shades of lamp) with only 500 points, deterministic methods (ConvOcc, IFNet) tend to fill the gaps between the points of the partial observation.

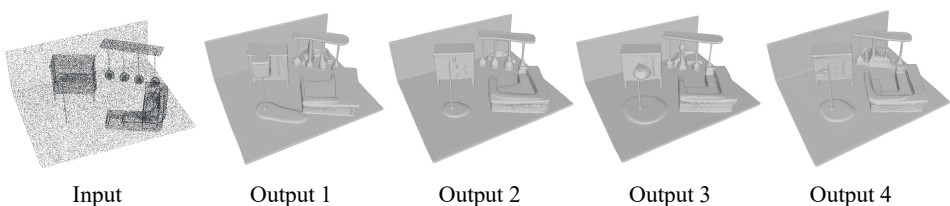

Figure 11: Qualitative results of cGCA tested on ShapeNet scenes where the input is non-ambiguous. The model is trained on completeness of 0.2.

Table 6: Quantitative results on cGCA trained on our ShapeNet dataset, but tested with non-ambiguous input. The first row (training) indicates the metrics evaluated on dataset created with same removal procedure as the corresponding training dataset. The second row (non-ambiguous) shows the metrics where the input from the test dataset is sampled very densely without any removal from the ground truth mesh with the same trained models as the first row. Both CD (quality, $\downarrow$) and TMD (diversity, $\uparrow$) in tables are multiplied by $10^4$.

| evaluation dataset | min. rate 0.2 | | | min. rate 0.5 | | | min. rate 0.8 | | |
|---|---|---|---|---|---|---|---|---|---|
| | min. CD | avg. CD | TMD | min. CD | avg. CD | TMD | min. CD | avg. CD | TMD |
| training | 2.80 | 3.88 | 10.07 | 1.16 | 1.49 | 3.91 | 0.69 | 0.87 | 3.05 |
| non-ambiguous | 2.46 | 3.58 | 8.72 | 0.73 | 0.87 | 2.91 | 0.61 | 0.71 | 2.72 |

This coincides with the result of Sec. F.1 that the reconstructions of our model is dependent on the multi-modality of plausible outcomes of input.

We also observe that the diversity of reconstructions is associated with the training data. Comparing the quantitative results on non-ambiguous input, the accuracy is low while the diversity is high when the model is trained on relatively incomplete data. We hypothesize that models trained with higher level of incompleteness are more likely to generate shapes from the input. However, Fig. 11 shows that while the completions of cGCA trained on highly incomplete data (min. rate 0.2) are diverse, they are quite plausible.

## G PROBABILISTIC SCENE COMPLETION ON REAL DATASET

We investigate how our model behaves on indoor real-world data. We test on the ScanNet indoor scene dataset (Dai et al. (2017a)), which is highly incomplete compared to other datasets, such as Matterport (Chang et al. (2017)). The input is sampled by collecting 500 points/m$^2$ from each scene. ScanNet dataset does not have a complete ground truth mesh, so we test our model trained on 3DFront (Fu et al. (2021)) dataset with completeness of 0.8. We align the walls of ScanNet data to xy-axis since the scenes of 3DFront are axis aligned. We compare our result with GCA (Zhang et al. (2021)).

The results are visualized in Fig. 12. Our model shows diverse results (e.g. chairs, closets) and generalizes well to the real data, which has significantly different statistics compared to the training data. Especially, the conditioned variant of our model shows better results by generalizing well to new data (e.g. tree), not found in 3DFront training dataset. We emphasize that our model is trained on unsigned distance fields and does not require the sliding-window technique, unlike the previous methods (Peng et al. (2020); Chibane et al. (2020a)).

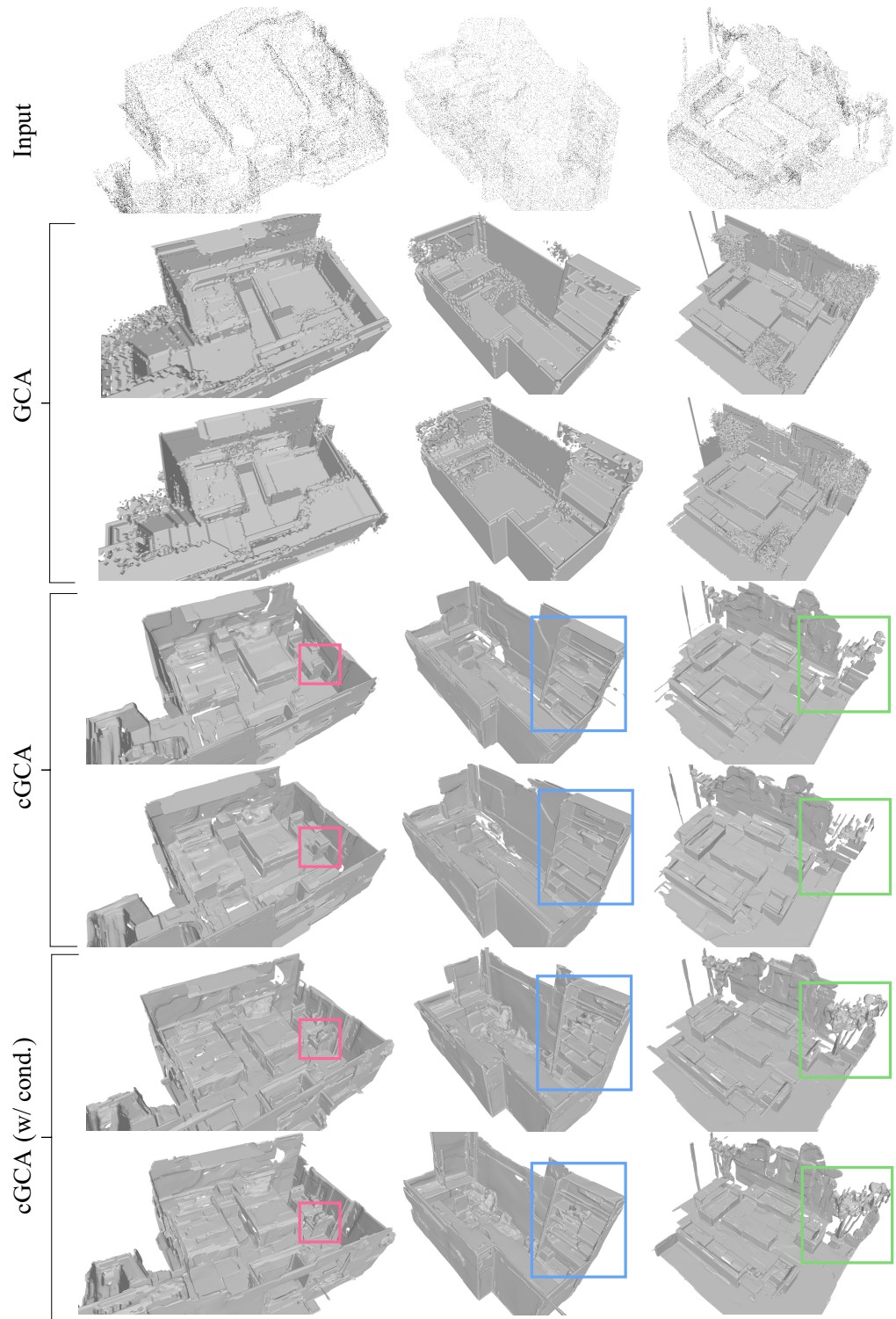

Figure 12: Qualitative results on ScanNet dataset with models trained on 3DFront dataset. Best viewed on screen. Multiple plausible reconstructions are shown (pink and blue box). cGCA (w/ cond.) shows better results for reconstructing a tree (green box) compared to that of vanilla cGCA, where a tree is never found in the training dataset. This allows us to infer that the conditioned variant tends to help generalize to unseen data better compared to the vanilla cGCA.

# H   ADDITIONAL RESULTS FOR SCENE COMPLETION

## H.1   ADDITIONAL RESULTS FOR SHAPENET SCENE

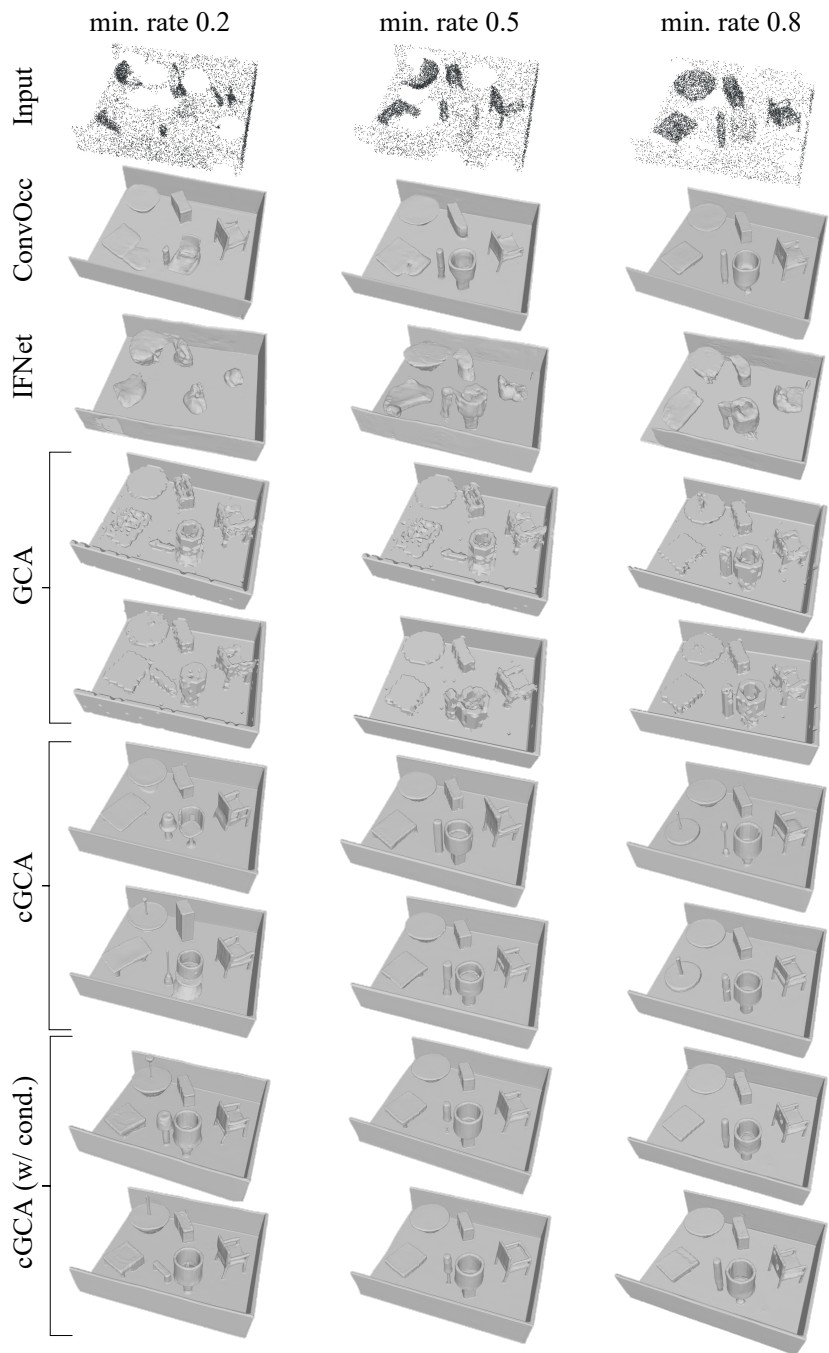

Figure 13:   Additional qualitative results on ShapeNet scene, with varying level of completeness. Best viewed on screen.

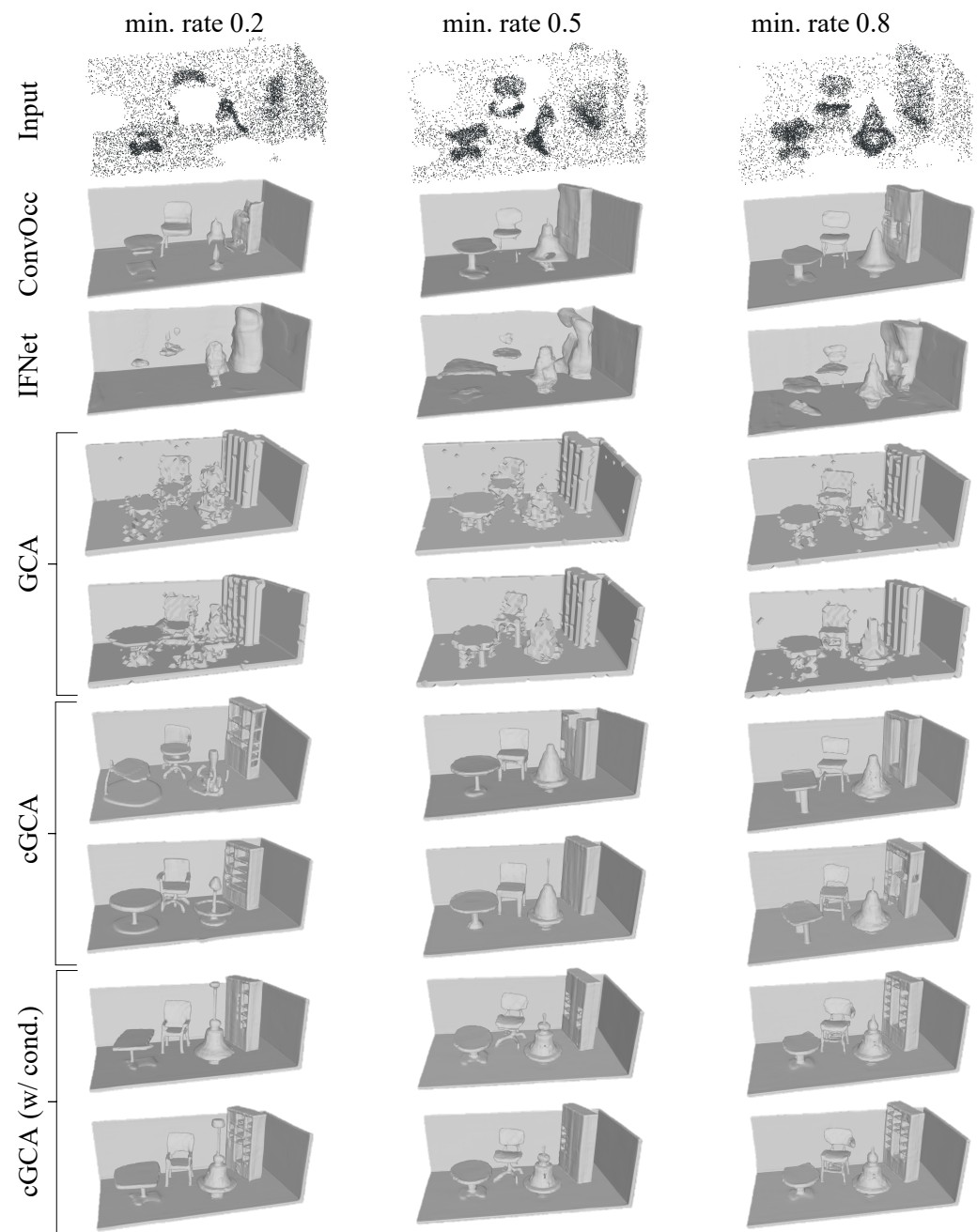

Figure 14: Additional qualitative results on ShapeNet scene, with varying level of completeness. Best viewed on screen.

## H.2 Additional Results for 3DFront

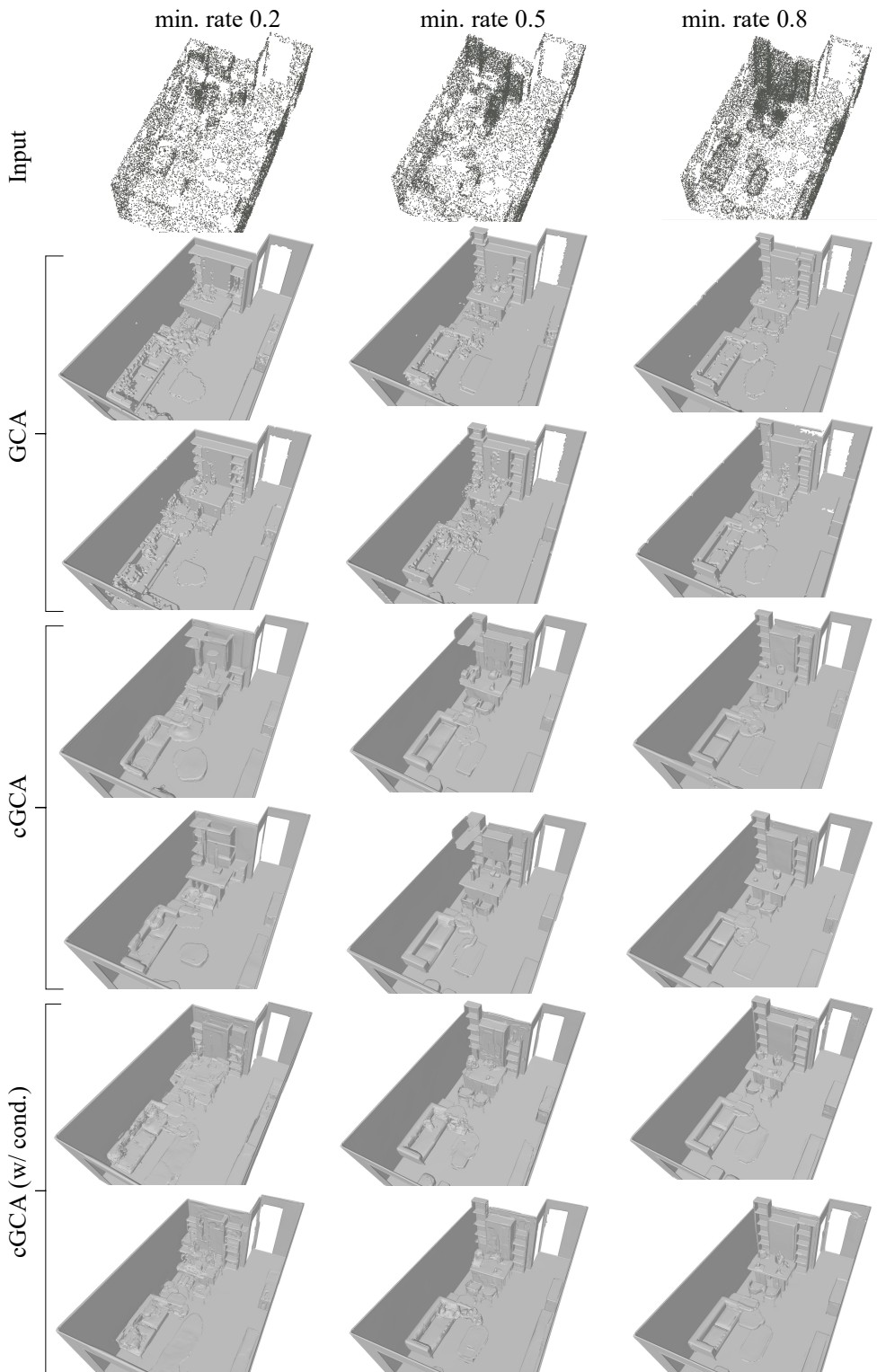

Figure 15: Additional qualitative results on 3DFront, with varying level of completeness. Best viewed on screen.

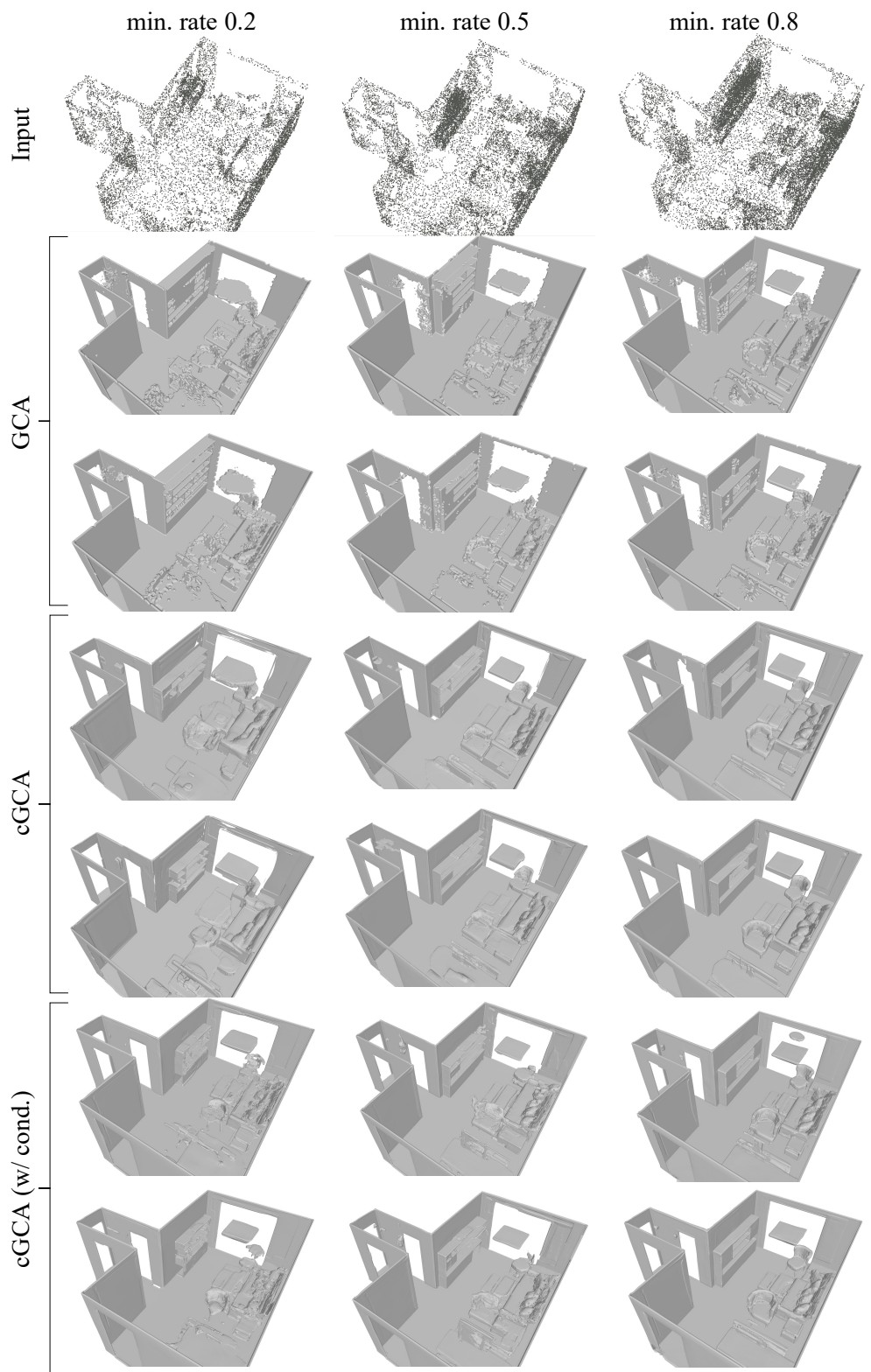

Figure 16: Additional qualitative results on 3DFront, with varying level of completeness. Best viewed on screen.

# I ADDITIONAL RESULTS FOR SINGLE OBJECT COMPLETION

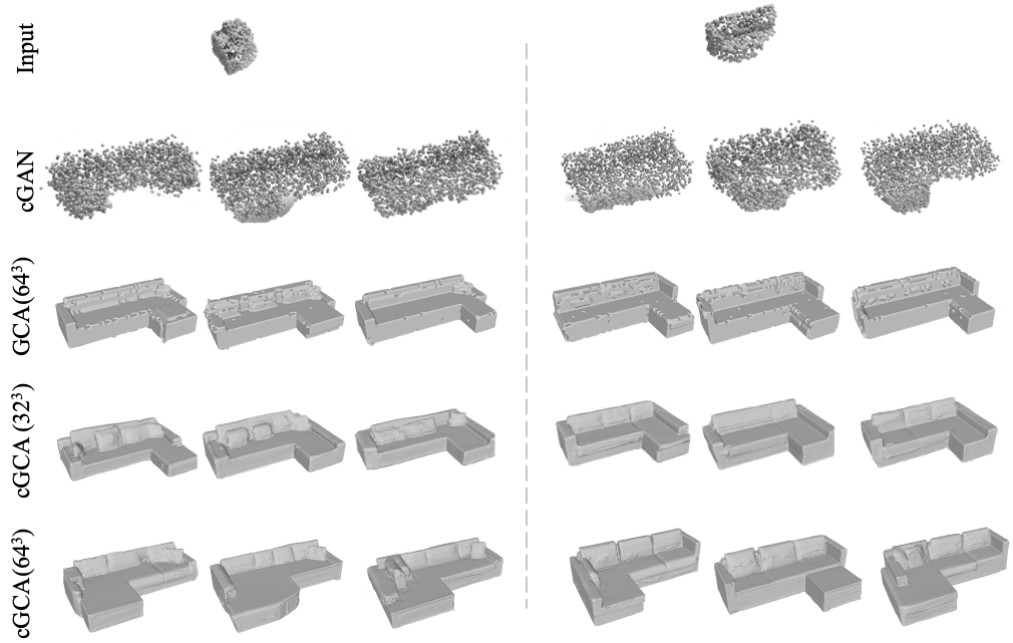

Figure 17: Additional qualitative results on ShapeNet sofa.

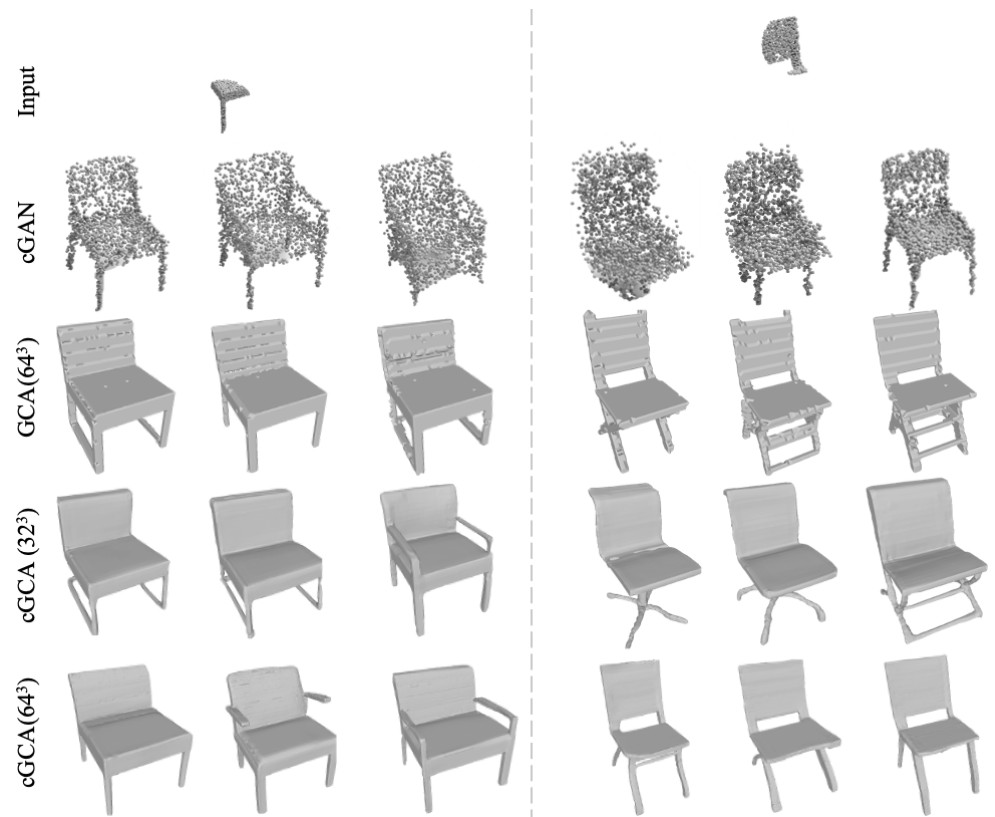

Figure 18: Additional qualitative results on ShapeNet chair.

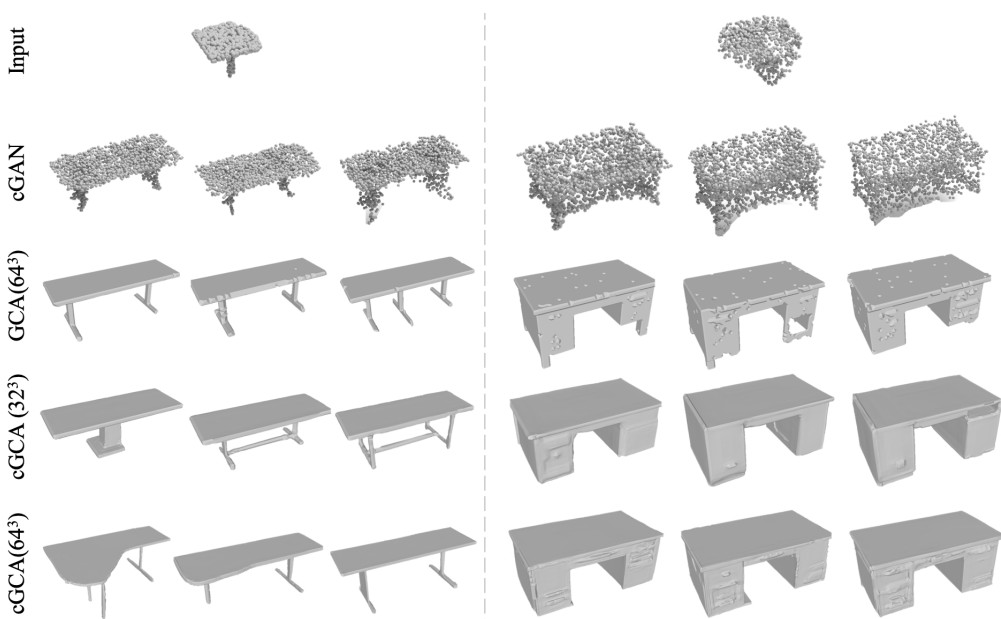

Figure 19: Additional qualitative results on ShapeNet table.

