# OpenReview forum: "Probabilistic Implicit Scene Completion"
_ICLR.cc/2022/Conference — ICLR 2022 Spotlight_

### Official Review · Reviewer_8mES · 2021-10-31

**Correctness:** 3
**Technical Novelty And Significance:** 3
**Empirical Novelty And Significance:** 2
**Recommendation:** 8
**Confidence:** 3

**Main Review:**

### Strengths
1. The authors tackle the problem of probabilistic scene completion with partial observations as input.
2. The authors apply the recent state-of-the-art neural implicit model for this probabilistic scene completion task, and show the ability of generating high-quality plausible shapes. The idea itself is simple and straightforward, but indeed effective.
3. The paper is well-written and easy to follow in most parts. Need to polish Sec 3.3 to make it more accessible.

### Weakness
**Why is the task itself important?**

This paper is trying to tackle “probabilistic scene completion”. Given partial point clouds, previous works (e.g. ConvONet, IF-Net) output a single most plausible shape. If you rerun the model with a different random seed, they usually also output a slightly different shape. This is also similar to my understanding of “probabilistic scene completion”. In the introduction, you did not explain why having multiple plausible shapes is interesting or important. Moreover, justify what makes your method really different from those previous works, if all methods can output slightly different outputs.

**The experimental section is convincing in general, but lacks some important experiments / explanation**

1. Why cGCA can produce less blurry shapes than the deterministic models as shown in Fig. 3? Please provide some explanations more than saying “coincides with the well-known phenomena”, since I don’t understand why different losses for the your generative model should perform better.
2. What is the model configuration that you use for the baseline? For example, for ConvOcc, do you use 3-plane model or grid model? What is the resolution for the planes or grids? What is the number of parameters in their model? How long do you train them? Same for IF-Net and your method. These network configurations can make a big difference for the reconstruction results, because a model with a lot of parameters and train for long time usually tends to produce better results.
3. For the mode seeking, why T’ is set to 5? An ablation study (e.g. T’ is set from 0 to 10) is necessary to show the effectiveness of mode seeking, and why you choose 5.
4. In ShapeNet Scene experiments, you use T=15 transitions for GCA and cGCA? For the single object completion, you use T=30 instead. Why you have different numbers for different tasks, and it is better to provide an ablation for the choices.
5. In Table 3, please give some explanations on 1) why (w/ cond.) does not really outperform the version w/o (2 out of 3 metrics are worse). 2) why 32^3 has really similar performance than 64^3.
6. In Table 4 in Appendix, the memory footprint for ConvOcc does not make sense. If I remember correctly, their memory usage is very constant but only the runtime increases linearly w.r.t. the number of crops because each crop is processed individually.

**What are the limitations of this paper?**

You mentioned in conclusion that the results are only evaluated with synthetic scene datasets. However, that is not really a limitation. It is always necessary to specify the real limitations of your approach. For instance, to obtain a complete shape voxel embeddings, previous methods like ConvOcc & IF-Net is pretty fast because you simply need to run encoder network containing some CNNs. In your method, you requires 15-30 transitions to obtain the final embeddings, which should be at least 15-30 slower, if I am not mistaken.




**Summary Of The Paper:**

This paper introduces continuous Generative Cellular Automata (cGCA) that is a generative model for continuous 3D reconstruction / shape completion. cGCA directly builds on top of GCA in the generation process, but instead applied the sparse voxel embeddings proposed in ConvONet and IF-Net to overcome the limitation of low resolution in GCA. Moreover, this paper adapts the infusion training strategy and also verifies the progressive generation is valid.

**Summary Of The Review:**

In general, this paper introduces an interesting way of combining neural implicit representations and generative models. The mathematical proofs seem to be sound and correct, but in the experimental part, there are some important experiments and explanations missing. I will change my score accordingly based on the reply from the authors.

Moreover, the previous work GCA has not made the code public. It would be great if the authors of cGCA can open source the code to benefit the community. This is another factor for me when making the decision.

---

> ### Author Response · Authors · 2021-11-18
> **Response to Reviewer 8mES (1)**
>
> We thank the reviewer for the constructive comments. Below, we address the concerns.
>
> &nbsp;&nbsp;
>
> > 1. This paper is trying to tackle “probabilistic scene completion”. Given partial point clouds, previous works (e.g. ConvONet, IF-Net) output a single most plausible shape. If you rerun the model with a different random seed, they usually also output a slightly different shape. This is also similar to my understanding of “probabilistic scene completion”. In the introduction, you did not explain why having multiple plausible shapes is interesting or important. Moreover, justify what makes your method really different from those previous works, if all methods can output slightly different outputs.
>
> We claim that the probabilistic formulation is essential for high-quality 3D shape completion, especially since there exists ambiguity for the solution. This is what we want to convey with the introduction of the main paper. Real-world scans are often highly irregular, due to complex occlusion and sensor noise. For example, given only the top part of a chair, there can be multiple plausible outcomes for the bottom part (four-legged, swivel, etc.). Therefore, the capability of processing the ambiguous scenes can largely benefit many tasks that require 3D scene understanding.  Previous works also acknowledge that 3D shape completion is an ill-posed problem and can have multiple plausible outcomes [1]. We need a solution to stably find one of the modes for high-fidelity completion. This is directly connected to producing probabilistic scene completion.
>
> In general, using a deterministic model to generate an outcome for multimodal distribution is known to generate a blurry outcome [4]. This is because deterministic models tend to converge into the average of the multiple possible outcomes. A similar trend is observable in our experiments, where the deterministic model (ConvOcc [2] or IFNet [3]) fails to reconstruct highly incomplete scans. To our understanding, ConvOcc [2] and IFNet [3] do not have any stochastic behavior in the model. Once a model is trained, it will always produce the same result. They originally demonstrate stable performance when the input is less ambiguous with uniformly sampled input points.
>
> On the other hand, our formulation can capture the large solution space of multimodal continuous 3D scenes, and demonstrate stable performance. The result also shows very different shapes, as shown in most of our figures. For example, the chairs in Fig. 5 are completely different ones, and the diversity cannot be achieved without a probabilistic formulation like ours. Given an ambiguous input, our model can generate multiple plausible results with a single training.
>
> &nbsp;&nbsp;
>
> > 2. Why cGCA can produce less blurry shapes than the deterministic models as shown in Fig. 3? Please provide some explanations more than saying “coincides with the well-known phenomena”, since I don’t understand why different losses for the your generative model should perform better.
>
> The phenomenon is related to modeling a multi-modal distribution with a generative model. For example, suppose we are to model one-dimensional data distribution, whose values are concentrated on either value 1 or 2. There are two dominant modes. A deterministic model trained with MSE loss can be interpreted as estimating a normal distribution (see [link](https://stats.stackexchange.com/a/288453)). If properly trained, it will have a mean at 1.5, which achieves the lowest training loss. However, a normal distribution with mean at 1.5 does not describe the multi-modal data well enough. This is the averaging effect by enforcing a single mode to underlying multi-modal data.
>
> The same phenomenon can be observed in high-dimensional data, which usually causes blurry outputs in image generation. A great example can be seen in Fig. 3 of GAN tutorial by Goodfellow [4], where a deterministic model produces a blurry prediction when learning a video sequence that has a multi-modal distribution.
>
> We can observe that a similar phenomenon also happens in the 3D completion task. For example, in the top row of Fig.3 ([highlighted image link](https://drive.google.com/file/d/19Xi-Ny9KNIxU-4jlnpgt7H3W8IP74rrP/view?usp=sharing)), the reconstructed parts of deterministic models tend to be blurry, such as the bottom of the lampshade (inside of the pink box) or top parts of the closet (inside of the blue box). In contrast, probabilistic models generate multiple shapes with crisp geometric details. As discussed, the deterministic models cannot express the multimodal distribution, and rather generate the average of multiple modes. Thus the shape will likely be not as detailed as the shapes of the individual modes. Further examples can be verified in Fig. 13 on page 27 and in Fig. 14 on page 28.

---

> > ### Author Response · Authors · 2021-11-18
> > **Response to Reviewer 8mES (2)**
> >
> > > 3. What is the model configuration that you use for the baseline? For example, for ConvOcc, do you use 3-plane model or grid model? What is the resolution for the planes or grids? What is the number of parameters in their model? How long do you train them? Same for IF-Net and your method. These network configurations can make a big difference for the reconstruction results, because a model with a lot of parameters and train for long time usually tends to produce better results.
> >
> > Due to the page limit, we included the model configuration for the baselines in Appendix A.3 in our original submission. We trained ConvOcc with a grid model of $64^3$ resolution, which produces the best result on the ShapeNet scene dataset in the original ConvOcc paper [2]. We trained IFNet with the ShapeNet128 model, as in the work of Siddiqui et al. [5]. We used the default hyperparameters and trained until validation IoU (or loss) converged. We also have tried other hyperparameters (e.g. learning rate, marching cubes thresholding parameters), but it did not observe any noticeable improvement.
> >
> > &nbsp; &nbsp;
> >
> > > 4. For the mode seeking, why T’ is set to 5? An ablation study (e.g. T’ is set from 0 to 10) is necessary to show the effectiveness of mode seeking, and why you choose 5.
> >
> > Thank you for the suggestion. It was also suggested by reviewer 2, and we have added an ablation study on mode seeking in Appendix E. Basically we compare the quality of reconstruction in response to the number of steps of mode seeking in Fig. 9. With a single step, we can remove all the voxels with low probability and greatly improve the final result. We chose T’=5 for all experiments to make sure that the shape reaches a stable mode of the probability distribution. We added a gif for the mesh reconstructions of each state in the [link](https://drive.google.com/file/d/1S8FS1Shmo5K1yOl-Yj3DPuW1FRbyEqPg/view?usp=sharing) to show the dramatic effect of the mode seeking steps.
> >
> > &nbsp;&nbsp;
> >
> > >5.  In ShapeNet Scene experiments, you use T=15 transitions for GCA and cGCA? For the single object completion, you use T=30 instead. Why you have different numbers for different tasks, and it is better to provide an ablation for the choices.
> >
> > The number of transition steps is the maximum distance we can expand from the initial incomplete shape. This is because GCA only can generate around the direct neighborhood of the occupied cells in a single inference step. We tried to set the transition steps to a minimum, since the longer the transition steps were, the longer the training/inference took. During the rebuttal period, we conducted an ablation study on the effect of transition steps by differing the transition steps trained with 15 transition steps. The below are the results.
> >
> > |               	| min. CD 	| avg. CD 	| TMD  	|
> > |---------------	|---------	|---------	|------	|
> > | ConvOcc       	|    1.33 	| -       	| -    	|
> > | cGCA (T = 15) 	|    1.16 	|    1.49 	| 4.92 	|
> > | cGCA (T = 20) 	|    1.19 	|    1.56 	| 4.29 	|
> > | cGCA (T = 25) 	|    1.24 	|    1.61 	| 4.34 	|
> > | cGCA (T = 30) 	|    1.26 	|    1.64 	| 4.51 	|
> >
> > The accuracy decreases (higher CD) as we increase T more than the trained value. Nonetheless, the accuracy is still better than ConvOcc with even twice as many transitions as the model is trained. Thus, our model is reasonably robust against T.
> >
> > Note that GCA also included a similar ablation study regarding the training step T (Appendix E in GCA [6]) and similarly concluded that the model is fairly robust to the number of transition steps.
> >
> > &nbsp;&nbsp;
> >
> > [1] Wu et al., Multimodal Shape Completion via Conditional Generative Adversarial Networks, ECCV 2020
> > [2] Peng et al., Convolutional Occupancy Networks, ECCV 2020
> > [3] Chibane et al., Implicit Functions in Feature Space for 3D Shape Reconstruction and Completion, CVPR 2020
> > [4] Goodfellow, NIPS 2016 tutorial: Generative adversarial networks, Arxiv 2016
> > [5] Siddiqui et al., Retrieval-Fuse: Neural 3D Scene Reconstruction with a Database, ICCV 2021
> > [6] Zhang et al., Learning to Generate 3D Shapes with Generative Cellular Automata, ICLR 2021

---

> > > ### Author Response · Authors · 2021-11-18
> > > **Response to Reviewer 8mES (3)**
> > >
> > > > 6. In Table 3, please give some explanations on 1) why (w/ cond.) does not really outperform the version w/o (2 out of 3 metrics are worse). 2) why 32^3 has really similar performance than 64^3.
> > >
> > > 1) Our results show that the (w/ cond.) outperforms in UHD (fidelity), similar results in MMD (quality). In terms of TMD (diversity), cGCA (w/ cond.) can be less diverse since it preserves the original initial shape. For example, in Figure 14 min. rate 0.2 of page 28 ([highlighted image link](https://drive.google.com/file/d/1XNIt2bqrm6llqyQKJJF09cD1wFLd8-H5/view)), the input provides the top of the lamp. While the GCA and vanilla cGCA modify the top part of the lamp, cGCA (w/ cond.) faithfully preserves the given input. Quantitatively, this results in better accuracy, but lower diversity.
> > > We claim that the diversity metric (TMD) should be jointly considered with the accuracy metrics (CD or MMD + UHD). As an extreme example, if a model generates completely random outputs and ignores the input, it will achieve a very high TMD but low accuracy scores.
> > >
> > > 2) The voxel resolution determines the level of geometric detail that a latent code needs to represent. Local implicit representation individually encodes smaller patches into latent codes, instead of modeling the entire shape with a single latent code. As discussed in previous works (Deep local shapes [7], Local implicit grid representations [8]), the patches of 3d surfaces share geometric details at some scale. The optimal resolution of a voxel is dependent on data. For single objects in ShapeNet, 32^3 appears to be a good enough resolution, capturing geometry comparable to 64^3.
> > >
> > > &nbsp;&nbsp;
> > >
> > > > 7. In Table 4 in Appendix, the memory footprint for ConvOcc does not make sense. If I remember correctly, their memory usage is very constant but only the runtime increases linearly w.r.t. the number of crops because each crop is processed individually.
> > >
> > > This is because we did not use the sliding windows techniques for measuring memory footprint for ConvOcc. The original ConvOcc dissects the scene into smaller sections and completes them separately. However, we argue that the sliding window technique is inappropriate for scene completion. The technique assumes that the divisions contain all the information necessary to complete the geometry. However, incomplete scans of a single object might be added into different segments, or the model will be trained to ignore the context outside of the current segment. This leads to inconsistent learning, and the model will fail to complete a scene when testing. Additionally, the model cannot observe the context beyond the current segment during testing. Thus, we have measured the memory footprint for processing the entire scene as input and result in a large memory footprint. Nonetheless, our formulation can efficiently handle the large-scale scenes.
> > >
> > > &nbsp;&nbsp;
> > >
> > > > 8. You mentioned in conclusion that the results are only evaluated with synthetic scene datasets. However, that is not really a limitation. It is always necessary to specify the real limitations of your approach. For instance, to obtain a complete shape voxel embeddings, previous methods like ConvOcc & IF-Net is pretty fast because you simply need to run encoder network containing some CNNs. In your method, you requires 15-30 transitions to obtain the final embeddings, which should be at least 15-30 slower, if I am not mistaken.
> > >
> > > With our current version of code, which is not optimized yet, the runtime is roughly 1.86 seconds. This is tested for a batch size of 1 on the ShapeNet sofa dataset with 64^3 resolution, 35 transitions (30 for sampling, 5 for mode seeking). The latent codes can be further decoded into a continuous 3D geometry in about 0.28 seconds, thus the run time is roughly 2.24 seconds in total. Note that we are using a batch size of 1, which can be accelerated with a larger batch size and the runtime varies largely on the number of input points.
> > >
> > > Admittedly our approach is slower than previous works formulated with a single forward pass. Nonetheless, we reconstruct high-fidelity large-scale scenes and handle multi-modal distribution of continuous shape. This is beyond the scale of any of the previous approaches that deal with a fixed number of points or are confined within a predefined voxel resolution.
> > >
> > > In addition, we would like to mention that the proposed number of transitions is much smaller than other 3D diffusion-based models [9, 10]. Other diffusion-based models use 200 or 1000 transitions for completing a single object, while our method uses at most 35. This is the advantage of infusion-based probabilistic models over diffusion-based models. Infusion-based models are known to generate a sample with a smaller number of time steps than the diffusion-based models, as discussed in Section 3.4 of [11].

---

> > > > ### Author Response · Authors · 2021-11-18
> > > > **Response to Reviewer 8mES (4)**
> > > >
> > > > > 9. Moreover, the previous work GCA has not made the code public. It would be great if the authors of cGCA can open source the code to benefit the community. This is another factor for me when making the decision.
> > > >
> > > > We have added the code for GCA in our Github repository.
> > > >
> > > > &nbsp;&nbsp;
> > > >
> > > > [7] Chabra et al., Deep Local Shapes: Learning Local SDF Priors for Detailed 3D Reconstruction, ECCV 2020
> > > > [8] Jiang et al., Local Implicit Grid Representations for 3D Scenes, CVPR 2020
> > > > [9] Luo et al., Diffusion Probabilistic Models for 3D Point Cloud Generation, CVPR 2021
> > > > [10] Zhou et al., 3D Shape Generation and Completion through Point-Voxel Diffusion, ICCV 2021
> > > > [11] Bordes et al., Learning to Generate Samples from Noise through Infusion Training, ICLR 2017

---

> > > > > ### Comment · Reviewer_8mES · 2021-11-18
> > > > > **Response to the Authors' Response**
> > > > >
> > > > > Thanks for the very detailed response, you addressed most of my concerns and I am in general very convince. I will upgrade my score accordingly. There are only two small points:
> > > > >
> > > > > * I don't agree with your argument about "the sliding-window technique is inappropriate for scene completion". In the section 6 of the [supplementary material](http://www.cvlibs.net/publications/Peng2020ECCV_supplementary.pdf) of ConvOcc, during training, although their crop size for occupancy prediction is only 25x25x25, their input crop for point cloud conditioning is much larger (88x88x88). In this way, they can incorporate enough context information to perform 3D reconstruction. During inference, they also have the input crops overlapped with each other. I argue that such sliding-window approach is the correct way of scaling up to very large scenes (e.g. in Matterport3D, a building with multiple floors) with constant memory footprint. I will not ask for this experiment of cGCA on Matterport3D since I am pretty sure cGCA can be adapted and working for such large scenes in the sliding-window manner as well.
> > > > > * I still think that it would be nice if you add a small limitation section of cGCA for readers, hence they would not waste too much time trying to figure out themselves. For example, as you also mentioned, your method is slower than other deterministic models because you need multiple transitions. The runtime will further increase if you increase the grid resolution.
> > > > >
> > > > >
> > > > >
> > > > > Last but not the least, thanks for promising to release the code, I am looking forward to trying it out!

---

> > > > > > ### Author Response · Authors · 2021-11-20
> > > > > > **Response to Reviewer 8mES**
> > > > > >
> > > > > > Thanks for the suggestion! We have added a statement regarding the limitations of runtime in the manuscript.

---

### Official Review · Reviewer_pTw1 · 2021-11-02

**Correctness:** 3
**Technical Novelty And Significance:** 3
**Empirical Novelty And Significance:** 3
**Recommendation:** 8
**Confidence:** 4

**Main Review:**

Strengths

1. The main novelty in this paper is the combination of a discretized voxel representation and implicit function representation in scene completion. Although both of them has been exploited before, combining them for the task of scene completion is a very natural and elegant. The key idea of this paper is augmenting the voxel with features that represent detailed local structures.

2. The proposed PointNet based encoder and implicit function decoder is new and inspiring as an autoencoder for dense point clouds or surface.

3. This paper delivers proof and derivation for the continuous version of GCA. Although the main chunk is similar to that of GCA, extending it with local features is non-trivial.

Weakness

My main concern is the choice of GCA as the operation for the voxel stage. As far as I know(maybe I'm wrong), GCA is still new and not  widely tested. However, I think the autoencoder proposed in this paper is irrelavant to the specific voxel-based operation, thus can be easily combined with other 3D object completion method that use also voxel-based operation, such as those baselines compared in this paper. While as the discussion of the limitation of GCA is out of the scope of this paper, this is not a big issue. But I think maybe the author can focus on the idea of augmenting voxel-based representation with local features when presenting the paper.  For example, I think Sec.3.1 should be much more importance then Sec.3.3. Most of the derivations in Sec3.3 can be moved to Appendix. Again, this is my personal idea.

**Summary Of The Paper:**

This paper is about scene completion given partial observations. The main technical part is built upon on prior work Generative Celluar Automata.(Zhang et al, 2021). GCA can be considered by recurrently applying a uniform convolutional operation on the top of current state, similar to the diffusion process. This work augments GCA by introducing a latent code for each voxel, instead of only a single 0-1 occupancy value. This latent code is used to generate fine detail implicit surface by querying a coordinated-based MLP inside each voxel. By combining the neural implicit function representation and GCA, the proposed method, namely cGCA outperforms prior works in terms of both scalability and fidelity.

As both the initial state and final state of cGCA is voxel-based representation, while the scene is represented as a continuous function or dense point cloud, an encoder-decoder network is used to convert the scene into voxel representation and vice versa.


========
Dongsu Zhang and Changwoon Choi and Jeonghwan Kim and Young Min Kim, "Learning to Generate 3D Shapes with Generative Cellular Automata", ICLR 2021

**Summary Of The Review:**

From my own perspective, this paper is interesting and strong. Especially, the scene local feature autoencoder presented in this paper is inspiring. This paper presents a way towards large-scale high-resolution scene generation/completion. Although I think the structuring of the paper could be improved.

---

> ### Author Response · Authors · 2021-11-18
> **Response to Reviewer pTw1**
>
> We appreciate the reviewer for the constructive  comments. Below, we address the concerns.
>
> &nbsp;&nbsp;
>
> > 1. My main concern is the choice of GCA as the operation for the voxel stage. As far as I know(maybe I'm wrong), GCA is still new and not widely tested. However, I think the autoencoder proposed in this paper is irrelavant to the specific voxel-based operation, thus can be easily combined with other 3D object completion method that use also voxel-based operation, such as those baselines compared in this paper. While as the discussion of the limitation of GCA is out of the scope of this paper, this is not a big issue.
>
> We would like to emphasize that one of the main contributions of our work is that we are the first to tackle the problem of *probabilistic* implicit scene completion. There have been a few works (ConvOcc [1], IFNet [2]) on scene completion, and they also utilize implicit function with autoencoder to encode the continuous shapes. However, they are deterministic and result in blurry completions given ambiguous inputs (see Figure 10). On the other hand, we explicitly model the multimodal distribution and achieve the state-of-the-art result.
>
> The main reason we adopt GCA [3] is to formulate the multi-modal distribution with infusion-based training. Few recent works, such as cGAN [4], tackle the multi-modal shape completion. But the experiments are only conducted only on single shape completions with 2,048 points. GCA outperforms cGAN and is more scalable by employing sparse convolution and the local update rules of cellular automata. We further modify GCA to formulate probabilistic implicit scene completion, and the difference we made is summarized in Appendix D.
>
> &nbsp;&nbsp;
>
> > 2. But I think maybe the author can focus on the idea of augmenting voxel-based representation with local features when presenting the paper. For example, I think Sec.3.1 should be much more importance then Sec.3.3. Most of the derivations in Sec3.3 can be moved to Appendix. Again, this is my personal idea.
>
> We argue that Section 3.3 is an essential part of our work. As you mentioned, extending GCA with local features is non-trivial. The problem becomes more challenging when designing the training objective with local features. In our work, we show that the derived training objective maximizes the variational lower bound of log-likelihood of the data. We believe that this is a significant theoretical contribution compared to GCA. GCA has not clearly shown the relationship of training objectives (what we actually do) and the log-likelihood of data (what we aim to achieve).
>
> &nbsp;&nbsp;
>
> [1] Peng et al., Convolutional Occupancy Networks, ECCV 2020
> [2] Chibane et al., Implicit Functions in Feature Space for 3D Shape Reconstruction and Completion, CVPR 2020
> [3] Zhang et al., Learning to Generate 3D Shapes with Generative Cellular Automata, ICLR 2021
> [4] Wu et al., Multimodal Shape Completion via Conditional Generative Adversarial Networks, ECCV 2020

---

> > ### Comment · Reviewer_pTw1 · 2021-11-24
> > **Raise to Accept**
> >
> > I'm satisfied with the response. Besides, after reading the comments from other reviewers, and also the responses, I'm feeling quite positive about this paper.

---

### Official Review · Reviewer_iZhE · 2021-11-03

**Correctness:** 4
**Technical Novelty And Significance:** 3
**Empirical Novelty And Significance:** 3
**Recommendation:** 8
**Confidence:** 4

**Main Review:**

Strengths:
- Shape completion is an important feature AR/VR, robotics and other applications
- The method is able to produce various plausible alternatives for shape completion
- The surfaces are continuous and quantitatively more accurate than the voxelized versions


Weaknesses:
- The benefit in terms of accuracy compared to the voxelized versions is coming at a cost of lesser diversity. Especially the conditioning on the initial state which helps a lot on the accuracy decreases diversity.
- In Fig. 3 shape completions with various amounts of sparsity in the input data is shown. However, they are all in a fairly dense range. What I would be interested is how would this model behave if the input becomes really sparse. When will it break down or will it still produce something meaningful with almost no input? On the other side it would also be interesting to know what happens if the input is very dense and the completion result is not ambiguous. I feel these would be interesting evaluations to better understand the limitations and strengths of this approach.
- From reading the submission I was not able to understand how seams between the voxels are prevented. It seems that there is a risk that such seams could be visible? Would they become visible if the input was sparser?
- It seems like the fact that during inference 30 rounds of state transitions needs to be computed would mean this method is significantly slower than the baselines presented in Fig 3. What are the runtimes of the proposed approach?
- One line of works I feel could be added to the related work is 3D shape representations based on octrees. I feel this would be an alternative formulation to the sparse voxel embedding to get the resolution of the shape higher than coarse voxels.

**Summary Of The Paper:**

This paper proposes an approach for scene completion which allows for sampling plausible completions from a learnt probabilistic model. The idea is to learn the state transition of a Markov Chain and use this during inference. The proposed formulation is based upon the Generative Cellular Automata where the representation of the surface is a discrete occupancy grid. The extension proposed in this work is to instead of using just voxel occupancy probabilities for the voxels also learn a shape code which can be decoded into a continuous surface, which is named sparse voxel embedding. The state transition is learned using an adapted version of infusion training.

**Summary Of The Review:**

The paper presents a method which is able to produce multiple plausible shape completions of a sparse input. The main novelty is adding a shape code to the voxel which allows for decoding continuous surfaces. The training procedure of the state transition was adjusted accordingly. The results look promising quantitatively and qualitatively. The submission has a few weaknesses but I think the approach is novel enough and shows quantitative improvement.

---

> ### Author Response · Authors · 2021-11-18
> **Resonse to Reviewer iZhE (1)**
>
> We thank the reviewer for the constructive and encouraging comments. Below, we address the concerns.
>
> &nbsp;&nbsp;
>
> >1. The benefit in terms of accuracy compared to the voxelized versions is coming at a cost of lesser diversity. Especially the conditioning on the initial state which helps a lot on the accuracy decreases diversity.
>
> The multi-modal shape completion assumes multiple answers and it is not simple to make an assessment about the performance. When we generate completely random shapes, we can achieve high diversity regardless of how faithful the final reconstruction is. On the other hand, if the input is almost complete to start with, we should not achieve high diversity. Therefore one needs to jointly consider the diversity (TMD) and the accuracy (CD or MMD + UHD).
>
> That said, we assume that cGCA (w/ cond.) becomes less diverse since it does not change the original initial state. For example, in Figure 10, min. rate 0.2 of page 23 ([highlighted image link](https://drive.google.com/file/d/1XNIt2bqrm6llqyQKJJF09cD1wFLd8-H5/view)), the input is the top part of a lamp. While the GCA and vanilla cGCA modify the input geometry, cGCA (w/ cond.) faithfully preserves it. Quantitatively, this results in better accuracy, but lower diversity.
>
> &nbsp;&nbsp;
>
> >2. In Fig. 3 shape completions with various amounts of sparsity in the input data is shown. However, they are all in a fairly dense range. What I would be interested is how would this model behave if the input becomes really sparse. When will it break down or will it still produce something meaningful with almost no input? On the other side it would also be interesting to know what happens if the input is very dense and the completion result is not ambiguous. I feel these would be interesting evaluations to better understand the limitations and strengths of this approach.
>
> Thank you for the suggestion, and we included the results in Appendix F. We originally followed previous work [1] and sampled 10000 points to produce input. Additionally, we experimented with different input sparsity (500, 1000, 5000, 10000 points) and tested the performance of the models trained with 10000 points.
>
> We show that our model (especially w/ cond.) is very robust against sparsity, achieving the best accuracy at all times (Table 5). While the quality of reconstruction deteriorates given only 500 points (x20 sparser than the training data) with all approaches, the probabilistic approaches (GCA, cGCA) still tries to generate the learned shapes (e.g., shades of the lamp in Fig. 10). On the other hand, the deterministic models tend to simply fill the space between points. The results agree with our assumption that modeling multimodal distribution is crucial for generating high-quality shapes in ambiguous cases.
>
> We also experimented with non-ambiguous input. We were also curious about how our model behaves on non-ambiguous input trained with varying levels of completeness. For all approaches, the accuracy increases and the diversity decreases as the ambiguity disappears for all models (Table 6). This confirms the diversity of reconstructions is dependent on the ambiguity of input. Also, the lower the completeness for training data, the more likely it was to generate diverse (yet plausible) scenes.
>
> &nbsp;&nbsp;
>
> >3. From reading the submission I was not able to understand how seams between the voxels are prevented. It seems that there is a risk that such seams could be visible? Would they become visible if the input was sparser?
>
> We did not observe any seams on the boundary of voxels for all the experiments, even for the sparse input experiments conducted in Appendix F. When we decode the continuous shape from sparse voxel embedding, each point observes the hierarchical context that aggregates multi-level features with sparse convolutional networks (see $f_{\omega_1}$ in Figure 6 and Appendix A.1). Therefore each point is affected by the multi-scale interpolation of convolved features at the given location, rather than the single latent code in a specific voxel. This approach is similar to [2] whereas we use sparse convolution instead of dense one. We added a short statement about the interpolation in the main text when we refer to the hierarchical autoencoder in Appendix A.
>
> Additionally, we can create a single coherent continuous shape with the help of mode seeking steps. The mode seeking step eliminates the spurious samples with low probability, and the output shape stably converges to a single dominant mode. We added the visualization in Fig. 9. Also, we added a gif for the mesh reconstructions of each state in the [link](https://drive.google.com/file/d/1S8FS1Shmo5K1yOl-Yj3DPuW1FRbyEqPg/view).
>
> &nbsp;&nbsp;
>
> [1] Peng et al., Convolutional Occupancy Networks, ECCV 2020
> [2] Chibane et al., Implicit Functions in Feature Space for 3D Shape Reconstruction and Completion, CVPR 2020

---

> > ### Author Response · Authors · 2021-11-18
> > **Resonse to Reviewer iZhE (2)**
> >
> > >4. It seems like the fact that during inference 30 rounds of state transitions needs to be computed would mean this method is significantly slower than the baselines presented in Fig 3. What are the runtimes of the proposed approach?
> >
> > With our current version of code, which is not optimized yet, the runtime is roughly 1.86 seconds. This is tested for a batch size of 1 on the ShapeNet sofa dataset with 64^3 resolution, 35 transitions (30 for sampling, 5 for mode seeking). The latent codes can be further decoded into a continuous 3D geometry in about 0.28 seconds, thus the run time is roughly 2.24 seconds in total. Note that we are using a batch size of 1, which can be accelerated with larger batch size and the runtime varies largely on the number of input points.
> >
> > Admittedly our approach is slower than previous works formulated with a single forward pass. Nonetheless, we reconstruct high-fidelity large-scale scenes, and handle multi-modal distribution of continuous shape. This is beyond the scale of any of the previous approaches that deal with a fixed number of points or are confined within a predefined voxel resolution.
> >
> > In addition, we would like to mention that the proposed number of transitions is much smaller than other 3D diffusion-based models [3, 4]. Other diffusion-based models use 200 or 1000 transitions for completing a single object, while our method uses at most 35. This is the advantage of infusion-based probabilistic models over diffusion-based models. Infusion-based models are known to generate a sample with a smaller number of time steps than the diffusion-based models, as discussed in Section 3.4 of [5].
> >
> > &nbsp;&nbsp;
> >
> > >5. One line of works I feel could be added to the related work is 3D shape representations based on octrees. I feel this would be an alternative formulation to the sparse voxel embedding to get the resolution of the shape higher than coarse voxels.
> >
> > We agree that we can handle high-resolution shapes with octrees, such as octree generating networks [6] or hierarchical implicit functions [7]. But we leave it for future work.
> >
> > &nbsp;&nbsp;
> >
> > [3] Luo et al., Diffusion Probabilistic Models for 3D Point Cloud Generation, CVPR 2021
> > [4] Zhou et al., 3D Shape Generation and Completion through Point-Voxel Diffusion, ICCV 2021
> > [5] Bordes et al., Learning to Generate Samples from Noise through Infusion Training, ICLR 2017
> > [6] Tatarchenko et al., Octree Generating Networks: Efficient Convolutional Architectures for High-resolution 3D Outputs, ICCV 2017
> > [7] Tang et al., OctField: Hierarchical Implicit Functions for 3D Modeling, NeurIPS 2021

---

> > > ### Comment · Reviewer_iZhE · 2021-11-27
> > > **Unchanged positive opinion about the submission**
> > >
> > > Thanks for the explanations, they answered my questions. After reading the authors answers and the other reviews I would like to keep my positive rating of this submission.

---

### Official Review · Reviewer_yCG6 · 2021-11-03

**Correctness:** 4
**Technical Novelty And Significance:** 3
**Empirical Novelty And Significance:** 3
**Recommendation:** 8
**Confidence:** 4

**Details Of Ethics Concerns:**

No concerns

**Main Review:**

Writing: The paper is well written and complete. The abstract lays out the structure of the paper and the text matches mostly up to the expectations set in the abstract and introduction. I did not find any obvious error. The contribution are explicitly stated and thereforce can be easily verified. The mathematical notation once introduced is static and easy to parse. The figures are sufficiently high resolution and demonstrative of the improvements due to the proposed method.

Implementation Details: I believe that the implementation is sufficiently detailed for a reader to be able to implement the algorithm - the previous models and their hyperparameters are described as are the datasets and their preprocessing steps.

Contribution: The paper extends GCA by using sparse voxel embeddings and a Distance Field method from Chibane et al. I think the contribution is pretty significant as the formulation under the new regime is vastly different from the old GCA in terms of implementation and engineering. The authors I think conclusively show that the new method is better performing on large scenes (Tables 1 and 2), while being competitive for single instances of 3D ojects (Table 3).

Comparisons: One could always ask from more comparisons, but I believe that the baselines are sufficient with two large scale methods - ConvOcc and IFNet, and comparison to the baseline GCA. Although it would be interesting to see how the authors managed to train GCA on such large scenes.

Proofs: The mathematics in the main paper was was checked and passed my review. However, I did not check the proofs in the supplementary.

Questions:
(Infusion Traninig): I am not entirely sure how the Infusion kernel is different from the infusion chain in GCA apart from the problem of having o_c inthe generative model itself instead of being defined on a grid. I would like to hear more explanation from the authors as the final update rule (pg 5, training procedure) is the same as the training procedure of GCA (Algorithm 1).

(Other methods): The authors do not make it clear how the other methods' performance was assessed - was it from the reported number or retrained with their preprocessing.

(Mode Seeking): The authors do not provide a model without mode seeking. It would be interesting to see the performance from such a model.






**Summary Of The Paper:**

Summary: The paper presents a shape completion method for large-scale 3d scenes. The method, called Continuous Generative Cellurlar Automata (cGCA)) is probabalistic in nature and produces multiple plausible completions for a single scene. The method employs leverages previous work in the form of key-buiilding blocks - Generative Cellular Automata, sparse voxel embeddings, and neural distance fields. Combining these approaches leads to impressive performance on recent 3D scene datasets - ShapeNet Scene and  3DFront. On ShapeNet the model outperforms Generative Cellular Automata (GCA) on reconstruction metrics, but not diversity metrics.

The main contribution of the paper to me is to realize that instead using a fixed grid like GCA to create voxel embeddings  leads to n^3 cost for the volume. In constrast, the authors use a sparse voxel embedding. They decode these embeddings with the method of  Chibane et al. to create a complete 3D shape. With this change, the method can scale to large scenes like from ShapeNet Scene and 3DFront, whereas GCA was limited to only small volumes(objects) of ShapeNet.



**Summary Of The Review:**

Summary of review: The paper is very well written with clear mathematics and clear delineation of contributions. The authors present a modification that makes older work scalable and show good results on multiple datasets, competing against the baseline method as well as other methods.

---

> ### Author Response · Authors · 2021-11-18
> **Response to Reviewer yCG6**
>
> We thank the reviewer for the effort in providing a detailed review of our work. We address the questions below:
>
> &nbsp;&nbsp;
>
> > 1. (Infusion Training): I am not entirely sure how the Infusion kernel is different from the infusion chain in GCA apart from the problem of having o_c in the generative model itself instead of being defined on a grid. I would like to hear more explanation from the authors as the final update rule (pg 5, training procedure) is the same as the training procedure of GCA (Algorithm 1).
>
> Both GCA and cGCA operate on sparse voxels, but cGCA extends the representation to contain the latent code instead of the simple binary occupancy in GCA. The latent code represents the local implicit representation, which can be decoded into a continuous shape. Our non-trivial contribution is to incorporate the latent code in the sparse formulation and derive the training algorithm for multi-modal distribution.
>
> We agree that the clear distinction between GCA and cGCA is necessary, and we added our explanation in Appendix C. We summarize the main points below.
>
> 1) Usage of latent codes. While the GCA operates on high-resolution voxels, it only generates binary occupancy (0 for empty, 1 for occupied) confined in the voxel resolution. On the other hand, cGCA uses local implicit latent code and generates continuous shape.
>
> 2) Processing disconnected shapes. In GCA, the proof of convergence (Proposition 1 in [1]) assumes (partial) connectivity of the shape. However, our derivation alleviates the connectivity assumption. Instead, we introduce an auxiliary function $G_x(s)$, which can deform the current input towards the desired output regardless of connectivity. Thus our formulation can train a generative model in a more general set-up.
>
> 3) Derivation of the training objective. GCA only provides the result and training procedure but does not provide the derivation of the training objective. However, our work provides the mathematical derivation of the training objective with proof that we are maximizing the lower bound of the log-likelihood of the distribution for the ground truth shape (Section 3.3 and Appendix B).
>
> The actual implementation is similar to Algorithm 1 in GCA [1], where the input buffer accelerates the training procedure and de-correlates the gradients of sequential models. The only difference between GCA and our model is the loss function (line 7 in Algorithm 1 [1]) which is replaced by Eq. 12 of our work, the new training objective proven to approximate the lower bound.
>
> &nbsp;&nbsp;
>
> >2. (Other methods): The authors do not make it clear how the other methods' performance was assessed - was it from the reported number or retrained with their preprocessing.
>
> All the results, including baselines, were re-trained for each dataset and each level of completeness, which was in Appendix A.3 in the initial submission. We realize that this is an important detail and added the statement in the main manuscript.
>
> &nbsp;&nbsp;
>
> >3. (Mode Seeking): The authors do not provide a model without mode seeking. It would be interesting to see the performance from such a model.
>
> We have added the ablation of mode seeking in Appendix E. Basically we compare the quality of reconstruction with a varying number of steps of mode seeking in Fig. 9. With a single step, we can remove all the voxels with low probability and greatly improve the final result. We chose T’=5 for all experiments to make sure that the shape reaches a stable mode of the probability distribution. We added a gif for the mesh reconstructions of each state in the [link](https://drive.google.com/file/d/1S8FS1Shmo5K1yOl-Yj3DPuW1FRbyEqPg/view?usp=sharing) to show the dramatic effect of the mode seeking steps.
>
> &nbsp;&nbsp;
>
> [1] Zhang et al., Learning to Generate 3D Shapes with Generative Cellular Automata, ICLR 2021

---

### Official Review · Reviewer_mswD · 2021-11-03

**Correctness:** 3
**Technical Novelty And Significance:** 3
**Empirical Novelty And Significance:** Not applicable
**Recommendation:** 8
**Confidence:** 3

**Main Review:**

Strengths:
1. The proposed method is clean and simple. The key component, cGCA, is a relatively straightforward extension of the prior work GCA to include latent embeddings and to use signed distance fields, but it is still brings enough novelty and contribution.
2. Being able to generate stochastic outputs is a significant advantage compared to prior work.
3. Under the experimental setup, the proposed method clearly outperforms prior work

Weaknesses:
1. Lack of experiments on real-world datasets: despite authors referred to this as one of the future directions, I think it is an important experiment to include as part of this submission. The noise statistics can be quite different, which could potentially break the proposed method.
2. My above concern is compounded by the fact that the authors were using a specific subsampling strategy to create synthetic inputs - i.e., each object has a guaranteed rate of surface points. This is significantly different from real-world pointcloud distributions. Are the prior work also re-trained specifically for this subsampled inputs? I would like to know how the method performs under uniform subsampling.
3. [Minor] In Equation (3), is $Q$ not defined?

**Summary Of The Paper:**

The submitted paper studies 3D scene geometry reconstruction and proposed a pipeline for predicting signed distance fields from incomplete pointcloud data. The pipeline consists of two major components: an autoencoder that maps between the latent space and the voxel representations, and a Continuous Generative Cellular Automata (cGCA) which progressively and stochastically generates more complete data at each step. The proposed method is trained and evaluated on several synthetic datasets, including ShapeNet Scene, 3DFront, as well as ShapeNet, with a controlled sub-sampled pointcloud as input (with guaranteed rate of points for each object). Under these settings, both quantitative and qualitative results outperformed prior work.


**Summary Of The Review:**

The authors' response and Appendix F & G address my concern regarding real-world experiments, so I am raising my ratings to "8: accept, good paper".

> [Original Review]
My current recommendation is "6: marginally above the acceptance threshold", given the simplicity of the method and its results on synthetic dataset. I am concerned about its actual performance on real-world data, and I am happy to increase my rating if the method demonstrates similar improvement over prior work on real-world datasets.

---

> ### Author Response · Authors · 2021-11-18
> **Response to Reviewer mswD**
>
> We thank the reviewer for the detailed comments. We address the replies below:
> &nbsp;&nbsp;
>
> >1. Lack of experiments on real-world datasets: despite authors referred to this as one of the future directions, I think it is an important experiment to include as part of this submission. The noise statistics can be quite different, which could potentially break the proposed method.
>
> During the discussion stage, we tested our method on the ScanNet [1] dataset, and the result is added in Appendix G. ScanNet is one of the widely used datasets for real-world indoor environments, acquired with real sensors. While our method was only trained on the 3DFront [2] dataset, our method nicely generalizes to the ScanNet dataset, given very different statistics compared to the 3DFront dataset. Our multi-modal generation creates diverse reconstructions (e.g. chair, closets) in Fig. 12 on page 26. Also, the conditioned variant of our model can reconstruct unseen objects such as trees better than the GCA or vanilla cGCA.
>
> &nbsp;&nbsp;
>
>
> >2. My above concern is compounded by the fact that the authors were using a specific subsampling strategy to create synthetic inputs - i.e., each object has a guaranteed rate of surface points. This is significantly different from real-world pointcloud distributions. Are the prior work also re-trained specifically for this subsampled inputs? I would like to know how the method performs under uniform subsampling.
>
> We have performed additional experiments regarding sparse inputs and non-ambiguous inputs in the Appendix. F. For sparse inputs, our method is the most robust method (especially the conditioned variant). Also, the diversity decreases with increasing accuracy as the input becomes less ambiguous. The same result is shown for non-ambiguous input (uniform sampling). The results show that our model is not just making diverse inputs, but the diversity of reconstruction is dependent on the ambiguity of the data.
>
> In the original submission, we maintain a fixed ratio of point clouds to form the task of scene completion. If there is no remaining portion of the shape, the methods will learn to generate an object without the relevant context, which is not what we intend to learn during training. This phenomenon will especially have a severe effect on the ShapeNet scene dataset, where the objects were generated at random.
> Our input point cloud reflects the scenario for scene completion for scans that suffer from complex occlusion, missing a large amount of continuous region. We demonstrate the performance with only 20% of the original shape and still produce reasonable scenes. On the other hand, previous works on shape completion, such as ConvOcc [3] uniformly subsamples 10,000 points. Our input, while we control the ratio of remaining shape, is more challenging and ambiguous than completing the uniformly sampled points, and better align to the formulation of the multi-modal generation.
>
> All the results, including baselines, were re-trained for each dataset and each level of completeness, which was in Appendix. A.3 in the initial submission. We realize that this is an important detail and added the statement in the main paper. For the ShapeNet scene dataset, as discussed in Appendix A.4, we uniformly sample 10,000 points after the iterative removal of points. The uniform sampling is exactly the same as that of ConvOcc, but ours is preceded by the iterative removal of points.
>
> &nbsp;&nbsp;
>
> >3. [Minor] In Equation (3), is Q not defined?
>
> Thank you for the correction. We have revised the manuscript.
>
> &nbsp;&nbsp;
>
> [1] Dai et al., Scannet: Richly-annotated 3D Reconstructions of Indoor Scenes, CVPR 2017
> [2] Fu et al., 3d-front: 3D Furnished Rooms with Layouts and Semantics, Arxiv 2020
> [3] Peng et al., Convolutional Occupancy Networks, ECCV 2020

---

> > ### Comment · Reviewer_mswD · 2021-11-23
> > **Raised my rating to "Accept"**
> >
> > Thanks for your response. Appendix F & G address my previous concern regarding real-world experiments, so I am raising my ratings to "8: accept, good paper".

---

### Author Response · Authors · 2021-11-18
**Common Response to All Reviewers**

We first would like to appreciate the thoughtful comments by the reviewers.


Here are the summary of the main changes and additional experiments conducted in the response of the reviews:
- We elaborate the difference of our proposed formulation compared to GCA in Appendix C.
- We added an ablation study of mode seeking steps in Appendix E.
- We examined the performance analysis under both sparse and non-ambiguous input in Appendix F.
- We provide results of our approach in real-world scans using the ScanNet dataset in Appendix G.


Also, we would like to emphasize that our work is not only presenting a method with superior performance but also is unique as it models the multi-modal distribution in the context of scene completion. Modeling the multi-modal distribution is crucial for high-quality generation in an under-constrained set-up, as a deterministic generative model would likely generate a blurry reconstruction given ambiguous input. Nonetheless, the field of learning multi-modal distribution is yet to be explored, to the best of our knowledge. Our contribution includes handling the high-dimensional solution space for a large-scale incomplete 3D scene.

---

### Decision · Program_Chairs · 2022-01-20

**Decision:**

Accept (Spotlight)

**Comment:**

This paper introduced a probabilistic extension to a pipeline for 3D scene geometry reconstruction from large-scale point clouds.
All reviewers recognized the significance of the proposed approach and praised the simplicity of deriving a probabilistic version of Generative Cellular Automata that performs well in a number of reconstruction benchmarks. Authors were responsive during rebuttal and managed to clarify the concerns raised about the limited scope of the experiments and certain parameters involved, and also raise one reviewer's scores.